# REMOVING INPUT FEATURES VIA A GENERATIVE MODEL TO EXPLAIN THEIR ATTRIBUTIONS TO AN IMAGE CLASSIFIER'S DECISIONS

## ABSTRACT

Interpretability methods often measure the contribution of an input feature to an image classifier's decisions by heuristically removing it via e.g. blurring, adding noise, or graying out, which often produce unrealistic, out-of-samples. Instead, we propose to integrate a generative inpainter into three representative attribution methods to remove an input feature. Compared to the original counterparts, our methods (1) generate more plausible counterfactual samples under the true data generating process; (2) are more robust to hyperparameter settings; and (3) localize objects more accurately. Our findings were consistent across both ImageNet and Places365 datasets and two different pairs of classifiers and inpainters.

## 1 INTRODUCTION

Explaining a classifier's outputs given a certain input is increasingly important, especially for life-critical applications (Doshi-Velez & Kim, 2017). A popular means for visually explaining an image classifier's decisions is an *attribution map* i.e. a heatmap that highlights the input pixels that are the evidence for and against the classification outputs (Montavon et al., 2018). To construct an attribution map, many methods approximate the attribution value of an input region by the classification probability change when that region is absent i.e. removed from the image. That is, most perturbation-based attribution methods implement the absence of an input feature by replacing it with (a) mean pixels; (b) random noise; or (c) blurred versions of the original content. While removing an input feature to measure its attribution is a principle method in causal reasoning, the existing removal (i.e. perturbation) techniques often produce out-of-distribution images (Fig. 1b,d) i.e. a type of adversarial example, which (1) we found to produce heatmaps that are sensitive to hyperparameter settings; and (2) questions the correctness of the heatmaps (Adebayo et al., 2018).

To combat these two issues, we propose to harness a state-of-the-art generative inpainting model (hereafter, an inpainter) to remove features from an input image and fill in with content that is plausible under the true data distribution. We test our approach on three representative attribution methods of Sliding-Patch (SP) (Zeiler & Fergus, 2014), LIME (Ribeiro et al., 2016), and Meaningful-Perturbation (MP) (Fong & Vedaldi, 2017) across two large-scale datasets of ImageNet (Russakovsky et al., 2015) and Places365 (Zhou et al., 2017). For each dataset, we use a separate pair of pre-trained image classifiers and inpainters. Our main findings are: [1]

1. Blurring or graying out the object in a photo yields images that are up to 3 times more recognizable to classifiers and remain more similar to the original photo (via MS-SSIM and LPIPS) than the images whose objects were removed via inpainting (Sec. 4.2).

2. Attribution methods with an inpainter produces (1) more plausible perturbation samples; (2) attribution maps that perform on par or better[2] than their original counterparts on three existing benchmarks: object localization, Insertion, and Deletion (see Sec. 4.4); (3) explanations that are more robust to hyperparameter changes (Sec. 4.3).

---

[1]All our code will be available on github.

[2]Our positive results here are new and in contrast with a previous result by Chang et al. (2019).

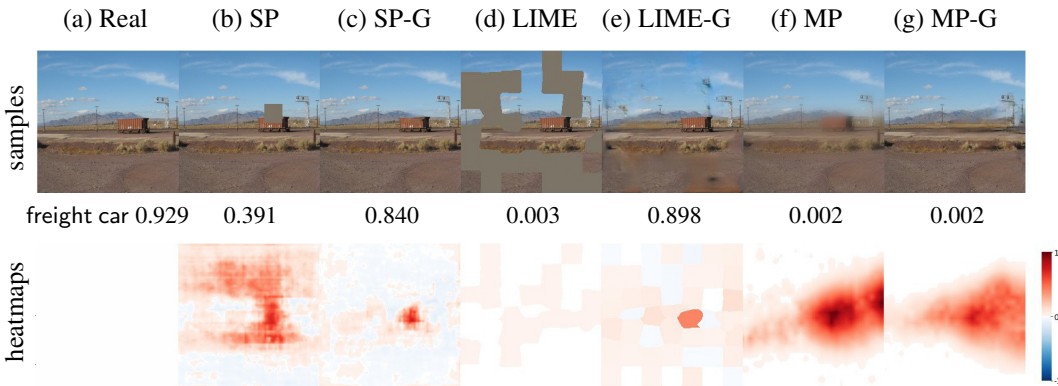

Figure 1: Three attribution methods, SP (Zeiler & Fergus, 2014), LIME (Ribeiro et al., 2016), and MP (Fong & Vedaldi, 2017), often produce unrealistic, out-of-distribution perturbation samples. **Top row:** SP slides a $29 \times 29$ gray patch across the image (b). LIME grays out a set of random superpixels (d). MP blurs out almost the entire image (f). In contrast, a learned inpainter integrated into these methods produces realistic samples for the same perturbation masks, here, completing the freight car (c), completing the background (e), and removing the car from the scene (g). Note that the freight car class probability is reduced by 57% (i.e. from 0.929 to 0.391) when only the top of the car is occluded (b). However, it is reduced by $\sim$100% down to 0.003 when the car is still present but the background is unnaturally masked out (d). Since the inpainted samples are more realistic, the probability drops are often less (c & e) and substantial only when the object is removed completely (g). **Bottom row:** the inpainted samples yield heatmaps that, in overall, outperform the original methods on the object localization task. Here, our heatmaps (SP-G, LIME-G, and MP-G) are less noisy and more focused on the object.

## 2 RELATED WORK

Attribution methods can be categorized into two main classes: (1) white-box and (2) black-box.

**White-box** Given the access to the network architecture and parameters, attribution maps can be constructed analytically from (a) the gradients of the output w.r.t. the input image (Simonyan et al., 2013), (b) the class activation map in fully-convolutional neural networks (Zhou et al., 2016), (c) both the gradients and activations (Selvaraju et al., 2017), or (d) the gradient times the input image (Shrikumar et al., 2017). However, these heatmaps can be too noisy to be human-interpretable because the gradients in the pixel space are often local. Importantly, some gradient-based attribution maps can be unfaithful explanations e.g. acting like edge detectors (Adebayo et al., 2018).

To make a gradient-based heatmap more robust and smooth, a number of methods essentially average out the resultant heatmaps across a large set of perturbed inputs that are created via (a) adding random noise to the input image (Smilkov et al., 2017; Fong & Vedaldi, 2017), (b) blurring or jittering the image (Fong & Vedaldi, 2017), or (c) linearly interpolating between the input and a reference "baseline" image (Sundararajan et al., 2017).

**Black-box** In addition to the white-box setting, perturbation techniques are even more important in approximating attributions under the black-box setting i.e. when we do not have access to the network parameters. Black-box methods often iteratively remove (i.e. occlude or perturb) an input region and take the average resultant classification probability change to be the attribution value for that region. While the idea is principle in causal reasoning, the physical interventions—taking an object out of a scene (revealing the content behind it) while keeping the rest of the scene intact—are impractical for most real-world applications.

**Feature removal** Therefore, the absence of an input region is often implemented by replacing it with (a) mean pixels (Zeiler & Fergus, 2014; Ribeiro et al., 2016); (b) random noise (Dabkowski & Gal, 2017; Lundberg & Lee, 2017); or (c) blurred versions of the original content (Fong & Vedaldi, 2017). However, these removal techniques often produce unrealistic, out-of-samples (Fig. 1), which raise huge concerns on the sensitivity and faithfulness of the explanations.

**Do explanations become more robust and faithful if input features are removed via a learned, natural image prior?** Here, we systematically study that question across three representative attribution methods: two black-box (i.e. SP and LIME) and one white-box (i.e. MP). We chose these three methods because they represent diverse approaches and perturb different types of input features: pixels (i.e. MP), superpixels (i.e. LIME); and square patches (i.e. SP). Similar to us, Chang et al. (2019) and Uzunova et al. (2019) also harnessed image generative models to remove input features. However, their findings were (1) both only within the MP framework; (2) either for grayscale, medical-image datasets (Uzunova et al., 2019) or based on unrealistic samples (see a comparison in Sec. 4.1). Furthermore, their results were not relevant to our question of whether integrating an inpainter helps attribution maps become more robust to hyperparameters. Note that Chang et al. (2019) found *negative* results for integrating the inpainter whereas our methods improved both the robustness of attribution maps (Sec. 4.3) and their ability to localize objects (Sec. 4.4).

**Counterfactual explanations** A task that is related but not the same as ours is to generate a textual explanation for why an image is predicted as class $c$ instead of a some other class $c'$ (Anne Hendricks et al., 2018). For visual explanations, Goyal et al. (2019) proposed to find a minimal input region such that when exchanged with another region in a reference image would change the classification for the original image into some target class. However, their counterfactual sample was generated by swapping patches between two images rather than by a generative model.

## 3 METHODS

### 3.1 DATASETS AND NETWORKS

**Classifiers** Our experiments were conducted separately with each of the two ResNet-50 image classifiers (He et al., 2016) that were pre-trained on the 1000-class ImageNet 2012 and Places365, respectively. The two models were officially released by the PyTorch (2019) model zoo and by the authors (CSAILVision, 2019), respectively.

**Datasets** We chose these two datasets because they are large-scale, natural-image sets and cover a wide range of images from object-centric (i.e. ImageNet) to scenery (i.e. Places365). While state-of-the-art image synthesis has advanced rapidly, unconditionally inpainting a large free-form mask in an arbitrary photo remains challenging (Yu et al., 2018a). Therefore, we ran our study on two subsets called ImageNet-S and Places365-S of the original validation sets of ImageNet and Places365, respectively, after filtering out semantically complex images. That is, we filtered out images that a YOLO-v3 object detector (Redmon & Farhadi, 2018) found more than one object. For ImageNet-S, we also filtered out images with more than one ImageNet bounding boxes. In total, ImageNet-S and Places365-S contains 15,082 and 13,864 images respectively (see Figs. S5 & S6 for example images).

**Inpainter** We used two TensorFlow DeepFill-v1 models pre-trained by Yu et al. (2018b) for ImageNet and Places365, respectively. DeepFill-v1 takes as input a color image and a binary mask, both at resolution $256 \times 256$, and outputs an inpainted image of the same size.

### 3.2 PROBLEM FORMULATION

Let $s : \mathbb{R}^{D \times D \times 3} \to \mathbb{R}$ be an image classifier that maps a square, color image $\boldsymbol{x}$ of spatial dimension $D \times D$ onto a softmax probability of a target class. An attribution map $\boldsymbol{A} \in [-1, 1]^{D \times D}$ associates each input pixel $\boldsymbol{x}_i$ to a scalar $\boldsymbol{A}_i \in [-1, 1]$ which indicates how much $\boldsymbol{x}_i$ contributes for or against the prediction score $s(\boldsymbol{x})$. We will describe below three different methods for generating attribution maps together with our three respective proposed variants which harness an inpainter.

### 3.3 SLIDING-PATCH

**SP** Zeiler & Fergus (2014) proposed to slide a gray, occlusion patch across the image and record the probability changes as attribution values in corresponding locations in the heatmap. That is, given a binary occlusion mask $m \in \{0, 1\}^{D \times D}$ (here, 1's inside the patch region and 0's otherwise) and a filler image $\boldsymbol{f} \in \mathbb{R}^{D \times D \times 3}$, a perturbed image $\bar{\boldsymbol{x}} \in \mathbb{R}^{D \times D \times 3}$ (see Fig. 1b) is given by:

$$\bar{\boldsymbol{x}} = \boldsymbol{x} \odot (1 - m) + \boldsymbol{f} \odot m \tag{1}$$

where $\odot$ denotes the Hadamard product and $\boldsymbol{f}$ is a zero image i.e. a gray image[3] before input pre-processing. For every pixel $\boldsymbol{x}_i$, one can generate a perturbation sample $\bar{\boldsymbol{x}}^i$ (i.e. by setting the patch center at $\boldsymbol{x}_i$) and compute the attribution value $\boldsymbol{A}_i = s(\boldsymbol{x}) - s(\bar{\boldsymbol{x}}^i)$. However, sliding the patch densely across the $224 \times 224$ input image of ResNet-50 is computationally expensive. Therefore, we chose a $29 \times 29$ occlusion patch size with stride 3, which yields a smaller heatmap $\boldsymbol{A}'$ of size $66 \times 66$. We bi-linearly upsampled $\boldsymbol{A}'$ to the image size to create the full-resolution $\boldsymbol{A}$. We implemented SP from scratch in PyTorch following a MATLAB implementation (MathWorks, 2019).

**SP-G** Note that the stride, size, and color of the SP sliding patch are three hyperparameters that are often chosen heuristically, and varying them can change the final heatmaps radically. To ameliorate the sensitivity to hyperparameter choices, we propose a variant called SP-G by only replacing the gray filler image of SP with the output image of an inpainter (described in Sec. 3.1) i.e. $\boldsymbol{f} = G(m, \boldsymbol{x})$ while keeping the rest of SP the same (Fig. 1b vs. c; top row). That is, at every location of the sliding window, SP-G queries the inpainter for content to fill in the window.

### 3.4 LOCAL INTERPRETABLE MODEL-AGNOSTIC EXPLANATIONS

**LIME** While SP occludes one square patch in the input at a time, LIME (Ribeiro et al., 2016) occludes a random-shaped region. The algorithm first segments the input image into $S$ non-overlapping superpixels (Achanta et al., 2012). Then, we generate a perturbed image $\bar{\boldsymbol{x}}$ by graying out a random set of superpixels among $2^S$ possible sets. That is, LIME follows Eq. 1 where the pixel-wise mask $m$ is derived from a random superpixel mask $m' \in \{0,1\}^S$. For each sample $\bar{\boldsymbol{x}}^i$, we measure the output score $s(\bar{\boldsymbol{x}}^i)$ and evenly distribute it among all occluded superpixels in $\bar{\boldsymbol{x}}^i$. Each superpixel's attribution is then inversely weighted by the $L_2$ distance $\|\boldsymbol{x} - \bar{\boldsymbol{x}}^i\|$ via an exponential kernel and then averaged out across $N$ samples. The resultant attribution $\boldsymbol{a}_k$ of a superpixel $k$ is finally assigned to all pixels in that group in the full-resolution heatmap $\boldsymbol{A}$.

In practice, Ribeiro et al. (2016) iteratively optimized for $\{\boldsymbol{a}_k\}_S$ via LASSO for 1000 steps to also maximize the number of zero attributions i.e. encouraging a simpler, sparse attribution map. We used the implementation provided by the authors (Ribeiro, 2019) and their default hyperparameters of $S = 50$ and $N = 1000$.

**LIME-G** While avoiding the bias given by the SP square patch, LIME perturbation samples remain unrealistic. Therefore, we propose LIME-G, a variant of LIME, by only changing the gray image $\boldsymbol{f}$ to a synthesized image $G(m, \boldsymbol{x})$ as in SP-G while keeping the rest of LIME unchanged.

### 3.5 MEANINGFUL PERTURBATION

**MP** Because SP and LIME gray out patches and superpixels in the input image, they generate unrealistic counterfactual samples and produce coarse heatmaps. To ameliorate these issues, Fong & Vedaldi (2017) proposed the MP algorithm i.e. learning a minimal, continuous mask $m \in [0,1]^{D \times D}$ that blurs out the input image in a way that would minimize the target-class probability. That is, they attempted to solve the following optimization problem:

$$m^* = \arg\min_m \lambda\|m\|_1 + s(\bar{\boldsymbol{x}}) \tag{2}$$

where $\bar{\boldsymbol{x}}$ is given by Eq. 1 but with $\boldsymbol{f} = B_\sigma(\boldsymbol{x})$ i.e. the input image blurred by a Gaussian kernel $B_\sigma(.)$ of radius $\sigma = 10$. Note that, in MP, the attribution map $\boldsymbol{A}$ is also the learned mask $m$.

However, solving Eq. 2 directly often yields heatmaps that are noisy and sensitive to hyperparameter changes. Therefore, Fong & Vedaldi (2017) only learned a small $28 \times 28$ mask and upsampled it to the image size in every optimization step. In addition, they also encouraged the mask to be smooth and robust to input changes by changing the objective function to the following:

$$m^* = \arg\min_m \lambda_1\|m\|_1 + \lambda_2 TV(m) + \mathbb{E}_{\tau \sim \mathcal{U}(0,4)}\big[s(\Phi(\bar{\boldsymbol{x}}, \tau))\big] \tag{3}$$

---

[3]The ImageNet mean pixel is gray (0.485, 0.456, 0.406).

where $TV(m) = \sum_i \|\nabla m_i\|_3^3$ i.e. a total-variation norm that acts as a smoothness prior over the mask. The third term is the expectation over a batch of randomly jittered versions of the blurred image $\bar{x}$. That is, $\Phi(.)$ is the jitter operator that translates an image $\bar{x}$ vertically or horizontally by $\tau$ pixels where $\tau$ is drawn from a discrete uniform distribution $\mathcal{U}(0, 4)$.

We randomly initialized the mask from a continuous uniform distribution $\mathcal{U}(0, 1)$ and minimize the objective function in Eq. 3 via gradient descent for 300 steps. Our MP implementation is in PyTorch and followed all the hyperparameters as described in Fong & Vedaldi (2017).

**MP-G**  We propose to create an MP variant called MP-G by only changing the filler image $f = B_\sigma(x)$ that is used in Eq. 1 to an inpainted image i.e. $f = G(m_b, x)$ where $m_b \in \{0, 1\}^{D \times D}$ is the result of binarizing $m$ at a threshold of $0.5$. Note that this binarization step is necessary because $G(.)$ expects a binary mask; however, we are still learning a continuous mask $m \in \mathbb{R}^{D \times D}$.

## 4 EXPERIMENTS AND RESULTS

### 4.1 INPAINTER FAILED TO SYNTHESIZE BACKGROUNDS GIVEN ONLY OBJECTS

Chang et al. (2019) proposed to find a minimal set of input pixels that would keep the classification outputs unchanged even when all other pixels are removed via an inpainter i.e. the "preservation" objective (Fong & Vedaldi, 2017). They used the same DeepFill-v1 inpainter as in our work; however, their "preservation" objective tends to force the inpainter to predict the missing background pixels conditioned on the remaining foreground object—a task that DeepFill-v1 was not trained to do and thus produced unrealistic samples as in Chang et al. (2019). In contrast, our MP-G method harnesses the dual "deletion" objective (Fong & Vedaldi, 2017) i.e. finding the smallest set of input pixels such that when inpainted would minimize the target-class probability—which intuitively asks the inpainter to replace the main object with some generated content similar to the real background.

To compare these two objectives, we randomly chose 50 validation-set images from 52 ImageNet bird classes and computed their segmentation masks via a pre-trained DeepLab model (Chen et al., 2017). We found that using the DeepFill-v1 to inpaint the foreground region (i.e. our "deletion" task) yields realistic samples where the object is removed. In contrast, using the inpainter to fill in the missing background area (i.e. the "preservation" task) yields unrealistic images whose backgrounds contain features (e.g. bird feathers or beaks) unnaturally pasted from the object (see a side-by-side comparison in Fig. S7). This result motivated us to integrate DeepFill-v1 into attribution methods and to study MP-G with the "deletion" objective which has not been explored in Chang et al. (2019).

### 4.2 INPAINTER IS EFFECTIVE IN REMOVING DISCRIMINATIVE FEATURES

While removing objects from an image via DeepFill-v1 qualitatively yields realistic samples, here, we quantitatively test how effective this strategy is in removing target-class discriminative features in comparison with three existing alternative filling methods: (1) zero pixels; (2) random noise; or (3) blurred versions of the original content.

Using the same procedure as described in Sec. 4.1, we randomly sampled 1000 bird images and segmented out the bird in each image. We filled in the object mask in each image via all four methods (Fig. 2) and compare the results (Table 1). Surprisingly, the blurred and grayed-out images are still correctly classified at 26.4% and 13.3% (Table 1), respectively, by a pre-trained Inception-v3 classifier (Szegedy et al., 2016). That is, these perturbed images still contain much discriminative features relevant to the target class. In contrast, only 8.9% of the inpainted images were correctly classified suggesting that the inpainter was more effective in removing the discriminative features.

After the main subject (here, birds) has been removed from an image, intuitively, one would expect the modified image to be perceptually different from the original image (where the bird exists). Here, we evaluate how each of the four in-filled images $\bar{x}$ (where the bird has been removed) is perceptually dissimilar to the original image $x$ by measuring the MS-SSIM and LPIPS (Zhang et al., 2018) scores between every pair $(x, \bar{x})$. Across both metrics, the inpainted images are consistently the most dissimilar from the real images compared to the blurred and grayed-out images. Note that in all three quantitative metrics, the inpainted images are the closest to the noise-filled images (Table 1d–e) despite being substantially more realistic (Fig. 2).

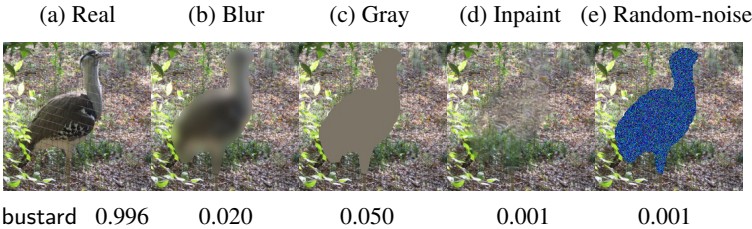

(a) Real  (b) Blur  (c) Gray  (d) Inpaint  (e) Random-noise

bustard  0.996  0.020  0.050  0.001  0.001

Figure 2: The results of filling the object mask in the real image (a) via four different filling methods. The shape of the bird is still visible even after blurring (b), graying out (c) or adding noise (e) to the bird region. The inpainter removes the bird and fills in with some realistic background content (d). Here, the bustard bird class probability for the inpainted image (d) is at random chance (0.001) suggesting that the Inception-v3 classifier does not detect any remaining bustard features.

| Metrics | Filling methods | | | | |
|---|---|---|---|---|---|
| | (a) Real | (b) Blur | (c) Gray | (d) Inpaint | (e) Random |
| Inception Accuracy | 92.30% | 26.40% | 13.30% | 8.90% | 4.40% |
| MS-SSIM (lower is better) | 1.000 | 0.941 | 0.731 | 0.707 | 0.692 |
| LPIPS     (higher is better) | 0.000 | 2.423 | 3.186 | 3.208 | 3.639 |

Table 1: Evaluation of four different filling methods on 1000 random bird images. The Inception-v3 accuracy scores (Salimans et al., 2016) suggest that inpainting the object mask (d) removes substantially more discriminative features relevant to the removed object compared to blurring (b) or graying out (c). Perceptually, the inpainted images are also more dissimilar to the corresponding real images according two similarity metrics: MS-SSIM and LPIPS (Zhang et al., 2018). See Fig. 2 for examples of all filling results.

### 4.3   Are generative attribution methods more robust to hyperparameters?

Perturbation-based attribution methods have many hyperparameters that are often heuristically tuned, which when varied can change the explanations radically. That sensitivity poses a huge challenge to (1) evaluating the explanations; and (2) building trust with end users (Doshi-Velez & Kim, 2017). Our hypothesis is that heuristically-perturbed samples are often far from the true data distribution and thus might contribute to the sensitivity of heatmaps to hyperparameters. Here, we test whether our attribution methods which incorporate an inpainter, i.e. SP-G, LIME-G, and MP-G (hereafter, G-methods), are more robust to hyperparameter changes than their original counterparts.

#### 4.3.1   Experiment setup

**Similarity metrics and Image sets**  Following Adebayo et al. (2018), we used three metrics from scikit-image (van der Walt et al., 2014) to measure the similarity of heatmaps: the structural similarity index (SSIM), the Pearson correlation of the histograms of gradients (HOGs), and the Spearman rank correlation. We upsampled all heatmaps to the full image size before feeding them into the similarity metrics. We performed the test on a set of 1000 random images from ImageNet-S and also replicated the experiments on Places365-S.

**SP sensitivity across patch sizes**  It remains a question how to choose the patch size in the SP algorithm because changing it can change the explanation radically (Zintgraf et al., 2017). Therefore, we compare the sensitivity of SP and SP-G when sweeping across 5 patch sizes $p \times p$ with stride 3 where $p \in \{5, 17, 29, 41, 53\}$. We chose this set to cover the common sizes that have been used in the literature. For each input image, we obtained $k = 5$ heatmaps (i.e. each corresponds to a patch size) and then measured the similarity among all $k(k-1)/2 = 10$ possible pairs.

**LIME sensitivity across random batches of samples**  LIME randomly samples $N$ perturbed images $\{\bar{x}^i\}_N$ and uses them to fit a heatmap. We found that the heatmap for an input can vary across different batches of random perturbation samples. Therefore, we compared the sensitivity of LIME and LIME-G across 5 random batches of $N = 500$ perturbation samples. That is, for each input

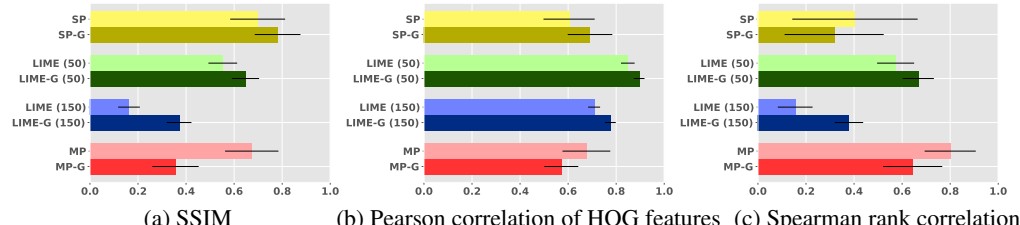

(a) SSIM            (b) Pearson correlation of HOG features  (c) Spearman rank correlation

Figure 3: Error plots for SSIM (a), Pearson correlation of HOG features (b), and Spearman rank correlation (c) scores obtained from 1000 random ImageNet-S images (higher is better). LIME-G is consistently more robust than LIME on both low and high resolutions i.e. $S \in \{50, 150\}$ (green and blue bars). The same trends were also observed on the Places365-S dataset (Fig. S4). The exact numbers are reported in Table S3.

image among the 1000, we generated $k = 5$ heatmaps and computed the similarity among all 10 possible pairs. We ran this experiment for a small and a large heatmap resolution i.e. two numbers of superpixels $S \in \{50, 150\}$ while keeping all other hyperparameters constant.

**MP sensitivity across random initializations** Similar to the LIME sensitivity experiment, here, we compared the variations of MP vs. MP-G heatmaps of size $28 \times 28$ across 5 random initializations. We re-ran each algorithm 5 times on each input image and computed the similarity among all 10 possible pairs. Here, all hyperparameters are the same as in Sec. 3.5.

### 4.3.2 RESULTS

First, we found that all 6 algorithms produce inconsistent explanations across the controlled hyperparameters (Fig. 3; all scores are below 1). That is, LIME and MP heatmaps can change as one simply changes the random seed! However, LIME-G is consistently more robust than LIME across all 3 metrics (Fig. 3) and across both superpixel settings. Across the patch sizes, SP-G is also consistently more robust than SP (Fig. 3a–b; light vs. dark yellow). SP-G and SP performed on par with high standard deviations under the Spearman rank correlation (Fig. 3c). Across the random initializations, MP-G is consistently less robust than MP (Fig. 3 light vs. dark red). The above trends were also observed on Places365-S (Fig. S4) suggesting that the findings are statistically significant.

### 4.4 ARE EXPLANATIONS BY GENERATIVE ATTRIBUTION METHODS MORE FAITHFUL?

While there are currently no established ground-truth datasets to evaluate an attribution map, prior research often assessed heatmaps via three proxy metrics: (1) the object localization task (Zhou et al., 2016); (2) Insertion and Deletion task (Petsiuk et al., 2018). Here, we ran all 6 algorithms on the ImageNet-S and Places365-S datasets using the default hyperparameters in Sec. 3. The heatmaps are then upsampled to the full image resolution for evaluation on all three benchmarks above.

**Object localization** Zhou et al. (2016) proposed to evaluate heatmaps by its ability to segment out objects in the ImageNet images, which often contain a single object of a known class. We applied the localization procedure in Fong & Vedaldi (2017) for the ImageNet-S dataset. That is, for each algorithm, we derived multiple bounding boxes per heatmap by thresholding it at different values of $t = \alpha \mu_{max}$, where $\mu_{max}$ is the maximum intensity in the heatmap and $\alpha \in [0 : 0.05 : 0.95]$. For each $\alpha$, we computed the Intersection over Union (IoU) score between a derived bounding box and the ImageNet ground-truth. The object localization error was calculated by thresholding each IoU score at $0.5$ and averaging them across the number of images. For each method, we chose the best $\alpha^*$ that yielded the lowest error on a held-out set of 1000 ImageNet-S images (Table 2).

We found that LIME-G and SP-G outperformed their respective counterparts while MP-G was on par with MP (Table 2). Among the 6 methods, LIME-G obtained the lowest error of 27.12%. For qualitative evaluation, Figs.S8–S10 show a set of heatmaps and derived bounding boxes for the cases where G-methods outperformed the original methods the most (in terms of IoU) and vice versa.

**Deletion and Insertion metrics** The Deletion metric measures the target-class probability changes as we gradually zero out the input pixels of the highest attributions one-by-one in the descending

| | baseline | SP | SP-G | LIME | LIME-G | MP | MP-G |
|---|---|---|---|---|---|---|---|
| $\alpha^*$ threshold | 0.5 | 0.05 | 0.05 | 0.15 | 0.15 | 0.20 | 0.50 |
| Error (%) | 38.56% | 39.17% | **34.87%** | 29.28% | **27.12%** | 29.98% | 30.43% |

Table 2: Localization errors (lower is better) for three G-methods and three originals on ImageNet-S. Naively taking the whole image as a bounding box yields an error of 38.56% (baseline). We also found SP-G to outperform SP on a replicated test with $53 \times 53$ patches (data not shown).

order. We evaluated all 6 methods via the code released by Petsiuk et al. (2018) on both the Deletion and also the related Insertion metric (Petsiuk et al., 2018; Samek et al., 2016). However, we did not find any consistent trends across both ImageNet-S and Places365-S datasets and both metrics. For example, SP-G consistently outperformed SP under the Deletion metric on both datasets, but not under the Insertion metric. See Sec. B for more details. Note that these metrics (1) assume that input pixels are *independent*; however, zero-ing out pixels one-by-one almost always yields unrealistic images; (2) the probability changes are computed from those unrealistic, out-of-distribution inputs and therefore can be misleading—the exact problem that our paper attempts to solve.

## 5 THE INNER-WORKINGS OF GENERATIVE ATTRIBUTION METHODS

Here, we used SP-G and LIME-G as the case studies to explain why our G-methods are both more (1) robust to hyperparameter changes (Sec. 4.3) and (2) accurate in localizing objects (Sec. 4.4).

### 5.1 MORE ACCURATE OBJECT LOCALIZATION: A CASE STUDY OF SP-G

We found that as the gray patch of SP is slid from left to right across the input image (Fig. 4a; top), the target-class probability gradually decreases and approaches 0 when the patch occludes most of the object (Fig. 4b; red line). Note that the probability can drop even when the patch is far outside the object region (Fig. 4b; red line within $[0, 24]$) due to SP unrealistic grayish samples. As the result, the probability distributions by SP often yield a large blob of high attributions around the object in the heatmap (Fig. 4a; top).

In contrast, the inpainted samples of SP-G often keep the probability variance low except when the patch overlaps with the object (Fig. 4b; blue vs. red), yielding heatmaps that are more localized towards the object (Fig. 4a; bottom). Across 1000 random ImageNet-S images, we found that the average probability change when the SP $53 \times 53$ patch is outside the object bounding box is $\sim 2.1$ times higher than that of SP-G (i.e. 0.09 vs. 0.04). In sum, our observations here are consistent with the findings that SP-G and LIME-G obtained lower localization errors than the original counterparts.

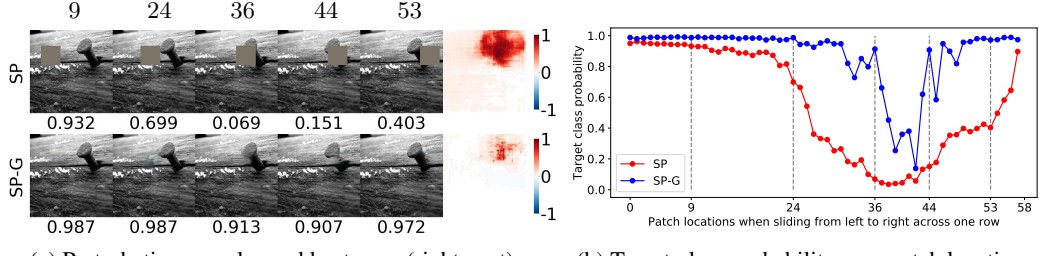

(a) Perturbation samples and heatmaps (rightmost)   (b) Target-class probability over patch locations

Figure 4: We ran SP and SP-G using a $53 \times 53$ patch on a nail class image. Here are the perturbation samples derived for both methods when the patch is slid horizontally across one row at 5 locations $\{9, 24, 36, 44, 53\}$ (a); and their respective target-class probability scores (b). SP-G samples are more realistic than those by SP and yielded a heatmap that localizes the object more accurately (a). That is, the probabilities for SP-G samples are more stable and only substantially drop when the patch covers the object (blue vs. red). See Fig. S11 for more examples.

## 5.2 MORE ROBUST HEATMAPS: A CASE STUDY OF LIME-G

Here, we provide insights for why LIME-G produced heatmaps that are more consistent than LIME across 5 random batches of samples (i.e. controlled, in practice, by 5 random seeds). We observed that the top-1 predicted labels of ∼20.5% of the LIME grayish perturbation samples (e.g. Fig. 5a) were from only three classes { jigsaw puzzle, maze, hen-of-the-wood } whereas the same top-1 label distribution for LIME-G samples was almost uniform (see Fig. S21 for more details). That is, across different objects, many LIME samples are still given the same label due to their similar grayish, puzzle-like patterns. Relatedly, we observed that a LIME perturbation sample is often given a near-zero probability score *regardless of what input feature is being masked out* (Fig. 5a). Therefore, when fitted to $N$ samples, where $N$ is often too small[4] w.r.t. the total $2^S$ possible samples, the heatmap appears random and changes upon a new set of random masks (Fig. 5b).

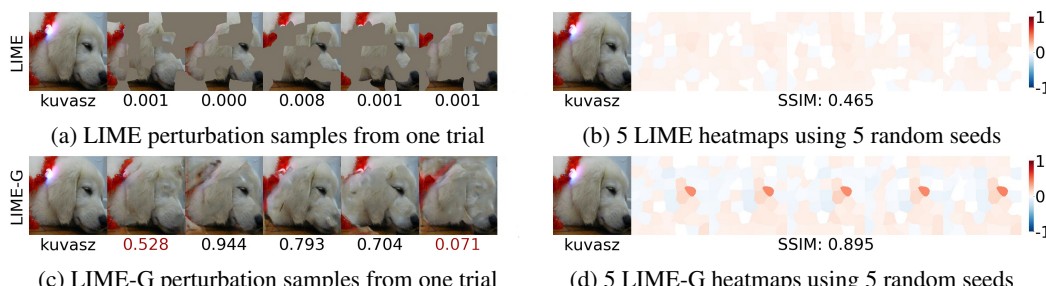

(a) LIME perturbation samples from one trial      (b) 5 LIME heatmaps using 5 random seeds

(c) LIME-G perturbation samples from one trial      (d) 5 LIME-G heatmaps using 5 random seeds

Figure 5: In Sec. 4.3, we compared the robustness of LIME vs. LIME-G heatmaps when running using 5 different random seeds. This is an example where LIME-G heatmaps are more consistent than LIME's (d vs. b). While LIME grayish samples (a) are given near-zero probabilities, LIME-G samples (here, inpainted using the same masks as those in the top row) are often given high probabilities except when the kuvasz dog's eye is removed (c). LIME-G consistently assign attributions to the dog's eye (d) while LIME heatmaps appear random (b). The top-1 predicted labels for 4 out of 5 LIME samples (a) here are paper towel.

In contrast, for LIME-G samples, the probabilities consistently drop when some discriminative features (e.g. the kuvasz dog's eye in Fig. 5c) are removed. This phenomenon yields heatmaps that are more consistently localized around the same input features across different random seeds (Fig. 5d). Our explanation also aligns with the finding that when the number of superpixels $S$ increases from 50 to 150 (while the sample size remains at $N = 500$), the sensitivity gap between LIME vs. LIME-G increases by ∼3 times (Fig. 3a; gap between green bars vs. gap between blue bars).

See Figs. S12–S19 for qualitative examples of when LIME-G is more robust than LIME and vice versa. Quantitatively, we found that the image distribution where LIME-G showed superior robustness over LIME *across all three similarity metrics* mostly contains images of scenes, close-up or tiny objects. In contrast, LIME is more robust than LIME-G on images of mostly birds and medium-sized objects (see Sec. C for more details).

## 6 DISCUSSION AND CONCLUSION

A hypothesis for why we did not find MP-G to outperform MP (Secs. 4.3 & 4.4) is that the original formulation (1) optimizes a tiny $28 \times 28$ mask; and (2) already strongly regularizes it to be sparse and smooth. A non-mutually exclusive hypothesis is that since MP operates in the pixel space, the learned masks can be of any shape. However, DeepFill-v1 was not designed for inpainting free-form masks (Yu et al., 2018a). We also found that a blur mask learned by MP often covers ∼99% of the input image. That is, the MP attribution map is explaining the target-class probability drops when the *entire image* is modified—an unrealistic causal intervention. In contrast, MP-G perturbations are more localized and only changes ∼40% of the input image (see Sec.A).

---

[4]Because sample size $N$ is often too small, for both LIME and LIME-G, the low-variance sample probability distribution (e.g. all near-zero scores in Fig. 5a) for an image is highly correlated with high across-random-seed sensitivity (Fig. S20; Pearson correlation $r = 0.856$).

In sum, integrating a state-of-the-art, unconditional inpainter into three existing attribution methods can yield visual explanations that (1) localize objects better; (2) are more robust to hyperparameter changes; and (3) are based on more plausible counterfactuals. Our results suggest that harnessing generative models to synthesize synthetic interventions (here, removal of input features) might be a promising direction for future causal explanation methods.

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

# APPENDIX

## A   MASKS LEARNED BY MP PERTURB ALMOST THE ENTIRE IMAGE

For ImageNet images where human-annotated bounding boxes only cover less than $50\%$ of the image, MP perturbs the entire image to generate its final attribution maps. Hence, the heatmap does not completely reflects the target object class as the drop in probability is due to the perturbation across the whole image (Fig. S1). On the other hand for MP-G, as discussed in Sec. 3.5, we binarize the mask during the optimization process and perturb only part of the image. In sum using MP-G, we perturb only $\sim 40\%$ and $\sim 39\%$ (Table S1) of the ImageNet-S and Places365-S images respectively.

| ImageNet BB | MP (ImageNet) | MP-G (ImageNet) | MP (Places365) | MP-G (Places365) |
|---|---|---|---|---|
| 20.28±12.01% | 99.30±1.24% | **40.22±23.11%** | 99.22±1.23% | **39.22±26.66%** |

Table S1: Average mean and deviation of the percentage of pixels perturbed by MP and MP-G for generating ImageNet-S and Places365-S heatmaps. For ImageNet-S, the ground truth bounding box (BB) covers a mean area of $20.28\%$ whereas, MP perturbs $\sim 99\%$ of the image and MP-G perturbs area closer to the actual ground truth distribution. See Fig.S1 for the qualitative represenation of these numbers.

| (a) Original | (b) MP heatmap | (c) MP-perturbed | (d) MP-G heatmap | (e) MP-G perturbed |
|---|---|---|---|---|

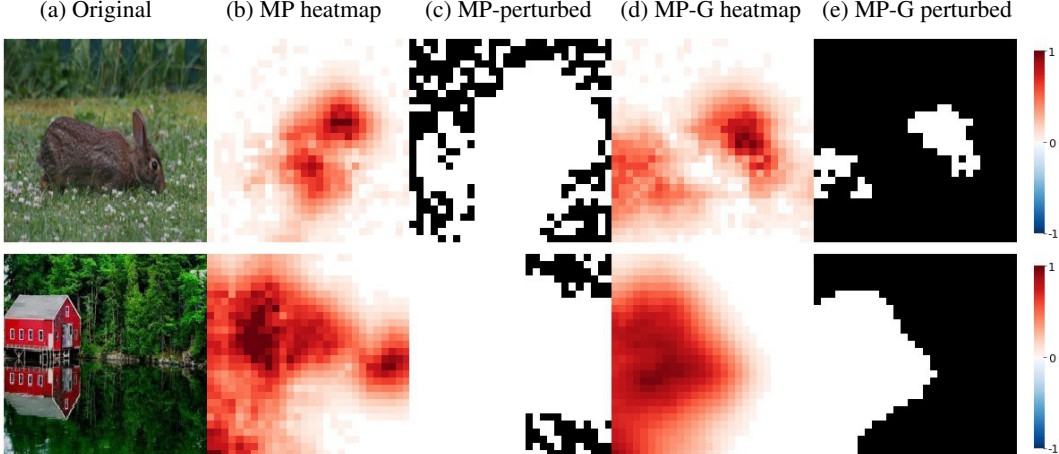

Figure S1: MP almost perturbs the entire image to generate its respective heatmap. **Top row:** An ImageNet-S image (a) and its respective MP (b) and MP-G (d) heatmaps. The pixels perturbed by MP (c) covers almost 99% of the pixels. In contrast, MP-G often perturbs ∼40% of the image (e). **Bottom row:** An example from the Places365-S dataset. MP-G just perturbs the area around the house (e) whereas, MP perturbs almost the entire image including the trees (c). These figures are qualitative evidence for the numbers in Table S1

.

# B  EVALUATION ON THE DELETION AND INSERTION METRICS

The deletion metric (Petsiuk et al., 2018) measures the target-class probability changes as we gradually zero out the input pixels of the highest attributions in the descending order. The idea is if the attribution values in a heatmap correctly reflect the discriminative power of the input pixels, knocking out the highest-attribution pixels should quickly cause the probability approach 0. However, this metric has two issues: (1) it makes an unrealistic assumption that input features are independent; (2) the probability changes may be due to the unrealistic, out-of-distribution inputs when some pixels are masked out. For completeness, we evaluated all 6 methods via the code released by Petsiuk et al. (2018) on the deletion metric and also its inverse version i.e. the insertion metric (Petsiuk et al., 2018) on both datasets of ImageNet-S and Places365-S. However, we did not find significant differences between the G-methods and their counterparts (Fig.S2).

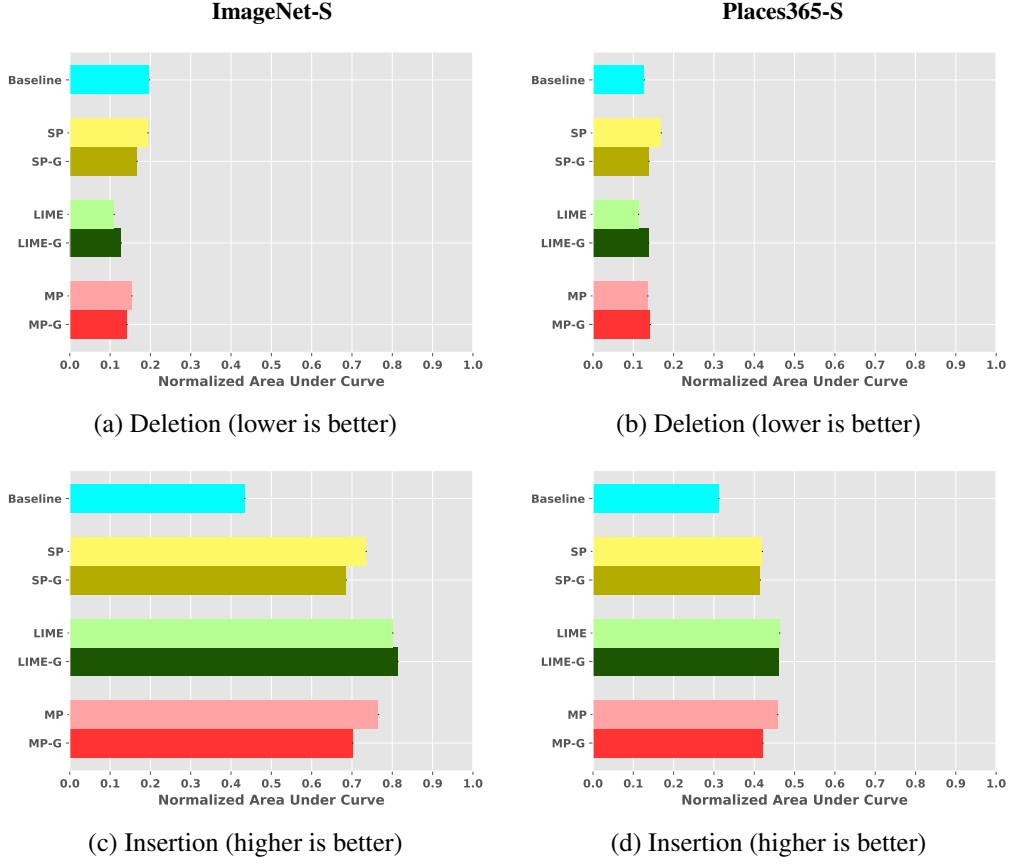

(a) Deletion (lower is better)      (b) Deletion (lower is better)

(c) Insertion (higher is better)      (d) Insertion (higher is better)

Figure S2: Error plots for the Deletion and Insertion metrics (Petsiuk et al., 2018) for all 6 attribution methods across two different datasets: ImageNet-S (a–c) and Places365-S (b–d). We used random noise heatmaps (i.e. having no information about the input image or the classifier) as the baseline method (cyan bars). We did not find any significant differences between G-methods and their respective counterparts across both two metrics and both datasets. Note that our Deletion and Insertion scores for the SP and LIME algorithms match closely to those in Petsiuk et al. (2018). See Table S2 for the exact numbers.

| | Dataset and Metric | | | |
| --- | --- | --- | --- | --- |
| | ImageNet-S | | Places365-S | |
| | Deletion | Insertion | Deletion | Insertion |
| Baseline | 0.197 | 0.433 | 0.126 | 0.311 |
| SP | 0.193 | 0.734 | 0.168 | 0.419 |
| SP-G | 0.166 | 0.685 | 0.138 | 0.414 |
| LIME | 0.109 | 0.800 | 0.112 | 0.462 |
| LIME-G | 0.127 | 0.813 | 0.137 | 0.459 |
| MP | 0.153 | 0.765 | 0.135 | 0.457 |
| MP-G | 0.141 | 0.701 | 0.141 | 0.421 |

Table S2: On the Deletion (lower is better) and Insertion (higher is better) metrics, we did not find any significant differences between the G-methods and their counterparts. See Fig. S2 for an error plot of this table.

## C  LIME-G IS MORE ROBUST THAN LIME ON IMAGES OF SCENES, CLOSE-UP AND TINY OBJECTS

We have shown that LIME-G is more robust than LIME consistently on all 3 different similarity metrics (see Sec. 4.3 in the main text). Here, we aim to understand the image distributions where LIME-G was more robust than LIME and vice versa.

For each of the three metrics, we computed a set of top-100 score differences between LIME-G vs. LIME. Interestingly, we found the intersection of the three sets contains images of mostly scenes, close-up or tiny objects (see Fig. S3). In contrast, the common set of images where LIME is more robust than LIME-G contains mostly birds and medium-sized objects. These image distributions intuitively align with the domains where DeepFill-v1 is capable of inpainting and suggest that the performance of G-methods can be improved further with class-conditional inpainters.

| Method | Similarity Metrics | | |
|---|---|---|---|
| | SSIM | Pearson correlation of HOGs | Spearman |
| SP | 0.698±0.114 | 0.604±0.106 | 0.404±0.261 |
| SP-G | **0.781±0.095** | **0.691±0.093** | 0.317±0.206 |
| LIME (50) | 0.553±0.060 | 0.848±0.028 | 0.573±0.077 |
| LIME-G (50) | **0.647±0.057** | **0.896±0.022** | **0.667±0.065** |
| LIME (150) | 0.163±0.045 | 0.708±0.025 | 0.155±0.072 |
| LIME-G (150) | **0.371±0.051** | **0.776±0.022** | **0.379±0.059** |
| MP | **0.673±0.111** | **0.676±0.099** | **0.800±0.106** |
| MP-G | 0.356±0.097 | 0.572±0.071 | 0.644±0.123 |

Table S3: The results in this table are the number forms of Fig.3. G-methods of SP and LIME are more robust to random parameters across different sensitivity metrics.

| Method | Similarity Metrics | | |
|---|---|---|---|
| | SSIM | Pearson correlation of HOGs | Spearman |
| SP | 0.577±0.177 | 0.674±0.073 | 0.452±0.288 |
| SP-G | **0.720±0.122** | **0.755±0.056** | **0.332±0.208** |
| LIME (50) | 0.392±0.074 | 0.802±0.036 | 0.594±0.078 |
| LIME-G (50) | **0.498±0.076** | **0.865±0.027** | **0.722±0.058** |
| LIME (150) | 0.118±0.046 | 0.701±0.026 | 0.201±0.071 |
| LIME-G (150) | **0.312±0.061** | **0.780±0.022** | **0.511±0.051** |
| MP | **0.703±0.112** | **0.736±0.097** | **0.846±0.124** |
| MP-G | 0.463±0.089 | 0.594±0.082 | 0.654±0.151 |

Table S4: The results in this table are the number forms of Fig.S4. The results follow the same trend as the ImageNet-S dataset.

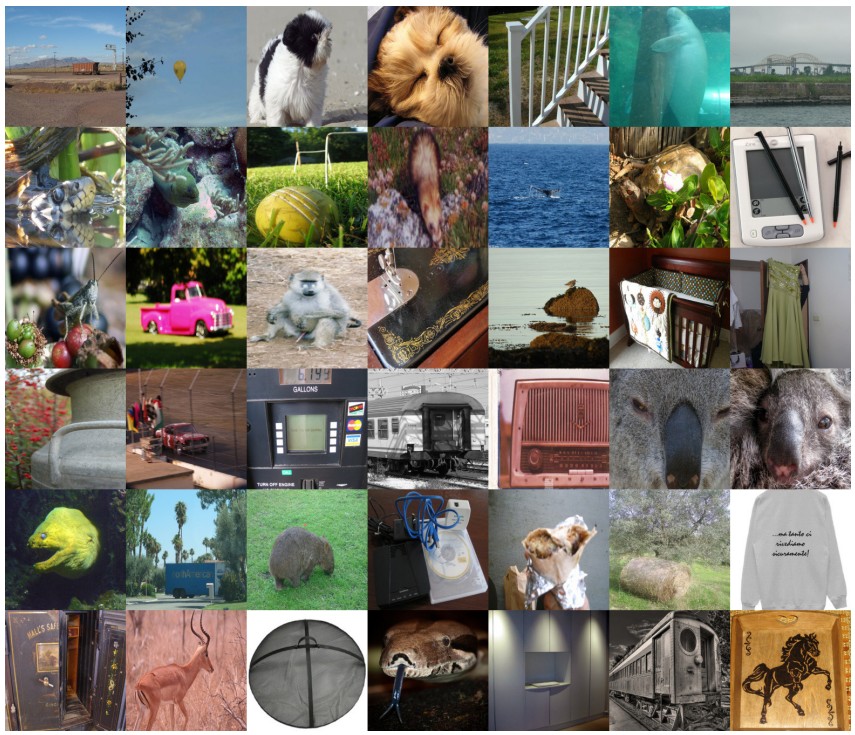

Images where LIME-G underperformed LIME across all three sensitivity metrics

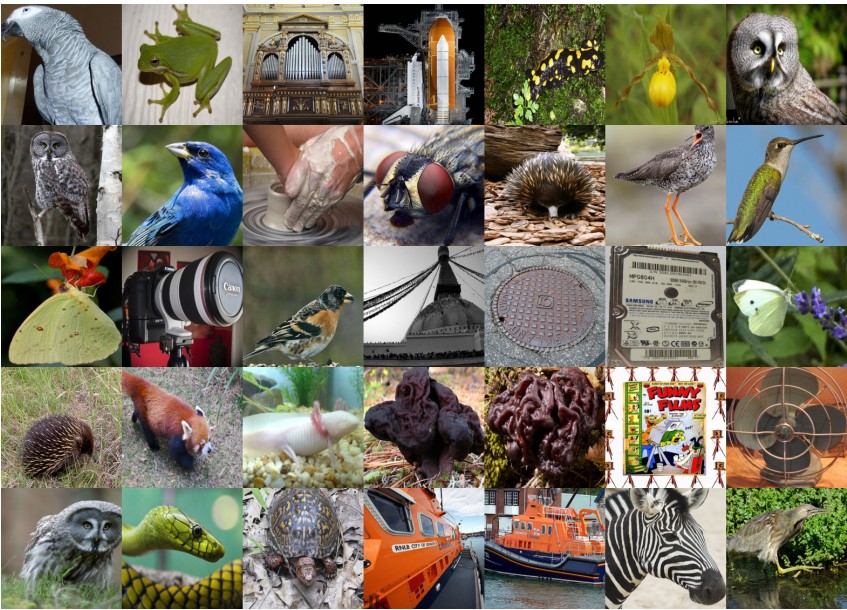

Images where LIME-G underperformed LIME across all three sensitivity metrics

Figure S3: Common images across all three metrics where LIME-G is consistently more robust than LIME (top) and vice versa (bottom). Interestingly, we found the intersection of the three sets contains images of mostly scenes, close-up or tiny objects (top). In contrast, the common set of images where LIME is more robust than LIME-G contains mostly birds and medium-sized objects (bottom).

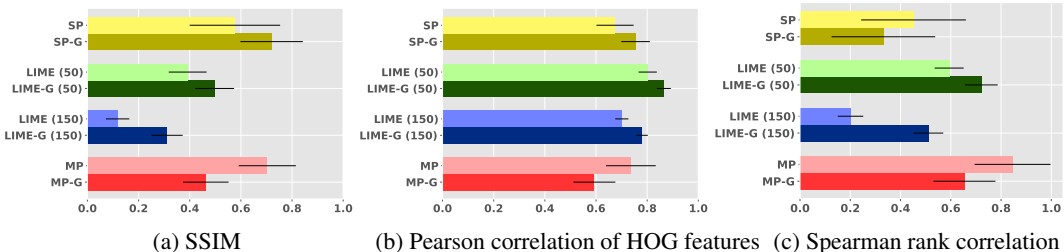

(a) SSIM    (b) Pearson correlation of HOG features  (c) Spearman rank correlation

Figure S4: Bar plots comparing the robustness (higher is better) of G-methods and their counter-parts when changing hyperparameters (described in Sec. 4.3) under three different similarity met-rics: SSIM (a), Pearson correlation of HOG features (b), and Spearman rank correlation (c). Each bar shows the mean and standard deviation similarity score across 1000 pairs of heatmaps, each produced for one random **Places365-S** image. SP-G is more robust than SP under SSIM (a) and Pearson correlation (b). In contrast, MP-G is consistently more sensitive than MP under all metrics (red bars are shorter than light-red bars). The exact numbers are reported in Table S4.

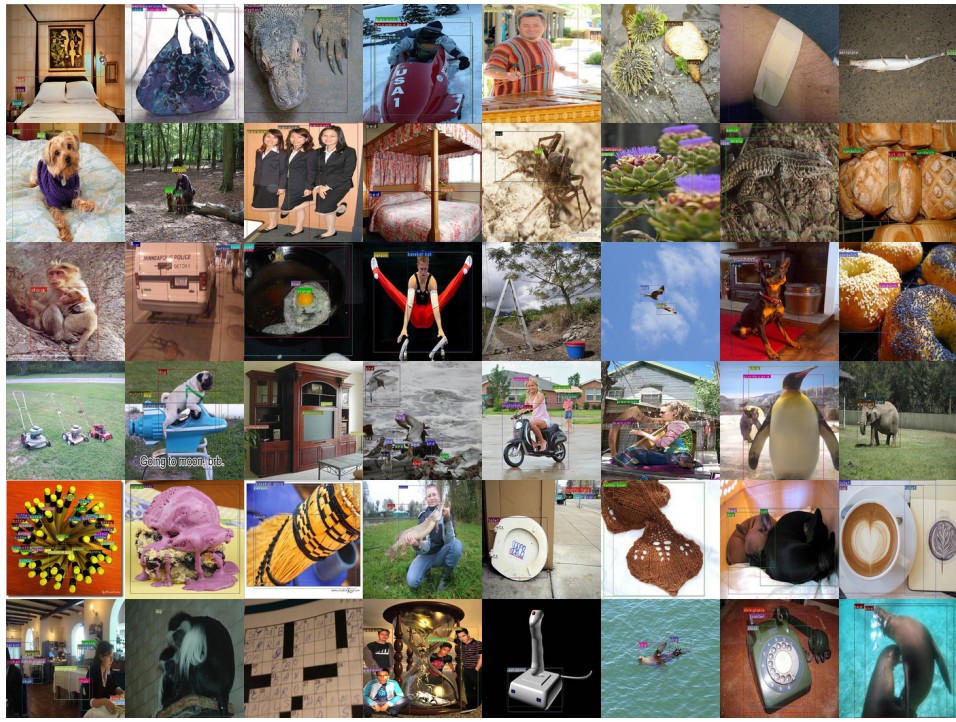

(a) ImageNet images where YOLO-v3 detects more than one objects and were not in ImageNet-S.

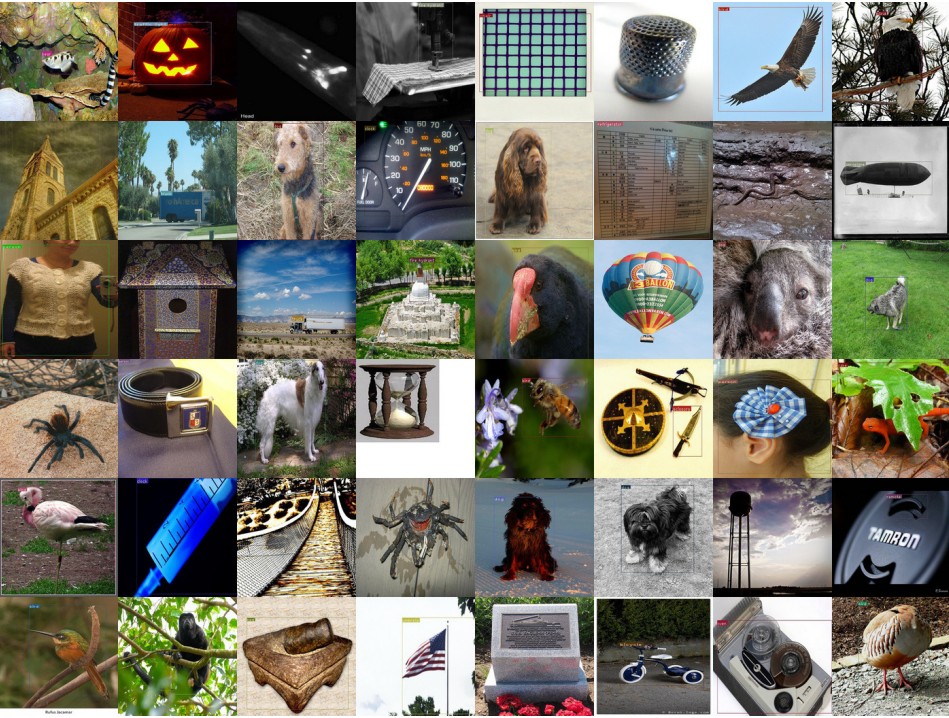

(b) Random images in ImageNet-S.

Figure S5: We filtered out images where YOLO-v3 detected more than one objects (a). The remaining images are used in ImageNet-S (b). We believe our observations on the ImageNet-S dataset carry over to the full ImageNet.

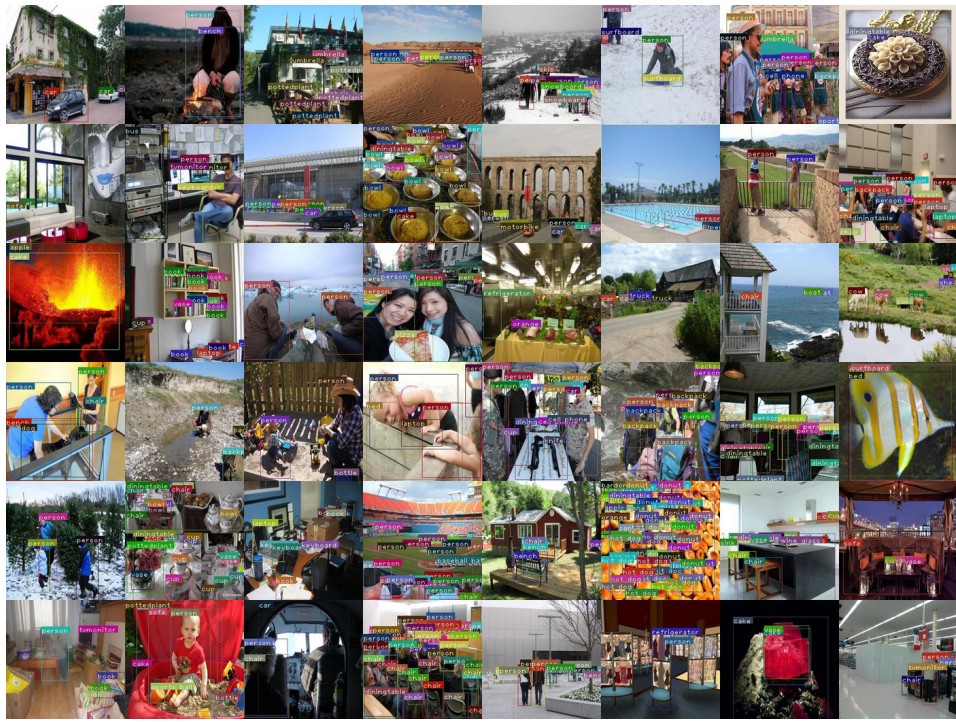

(a) Images in Places365 images that YOLO-v3 detected more than one objects and were not used in Places365-S.

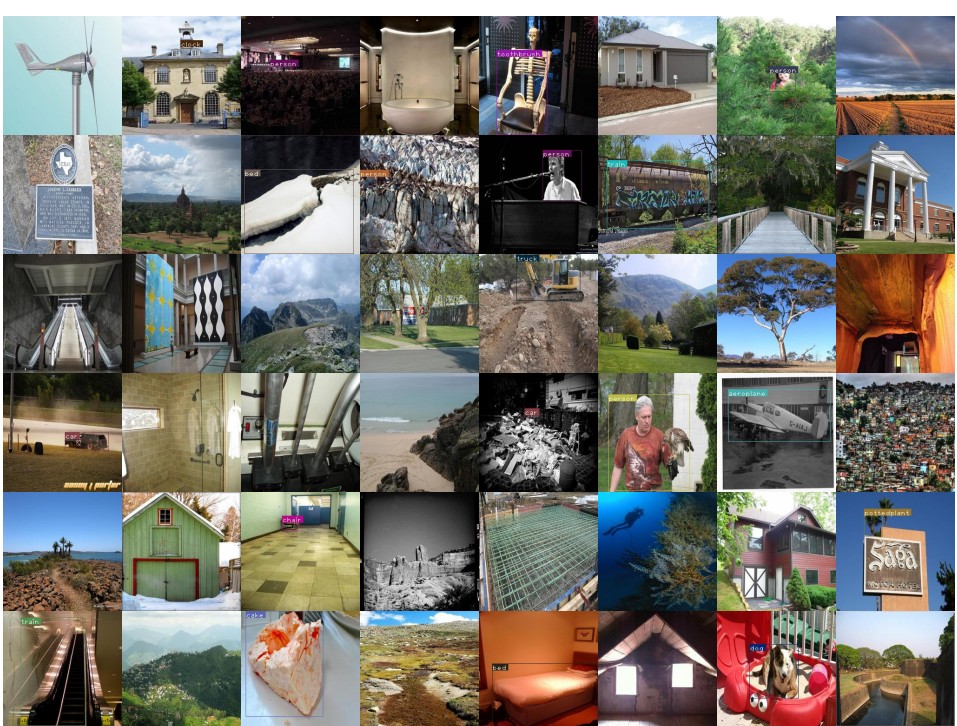

(b) Random Places365-S images.

Figure S6: Example images from Places365-S dataset (b). The images where YOLO-v3 detected more than one objects (a) are more cluttered compared to the images used in Places365-S (b).

(a) Real   (b) Mask   (c) Preserve   (d) Delete   (e) Real   (f) Mask   (g) Preserve   (h) Delete

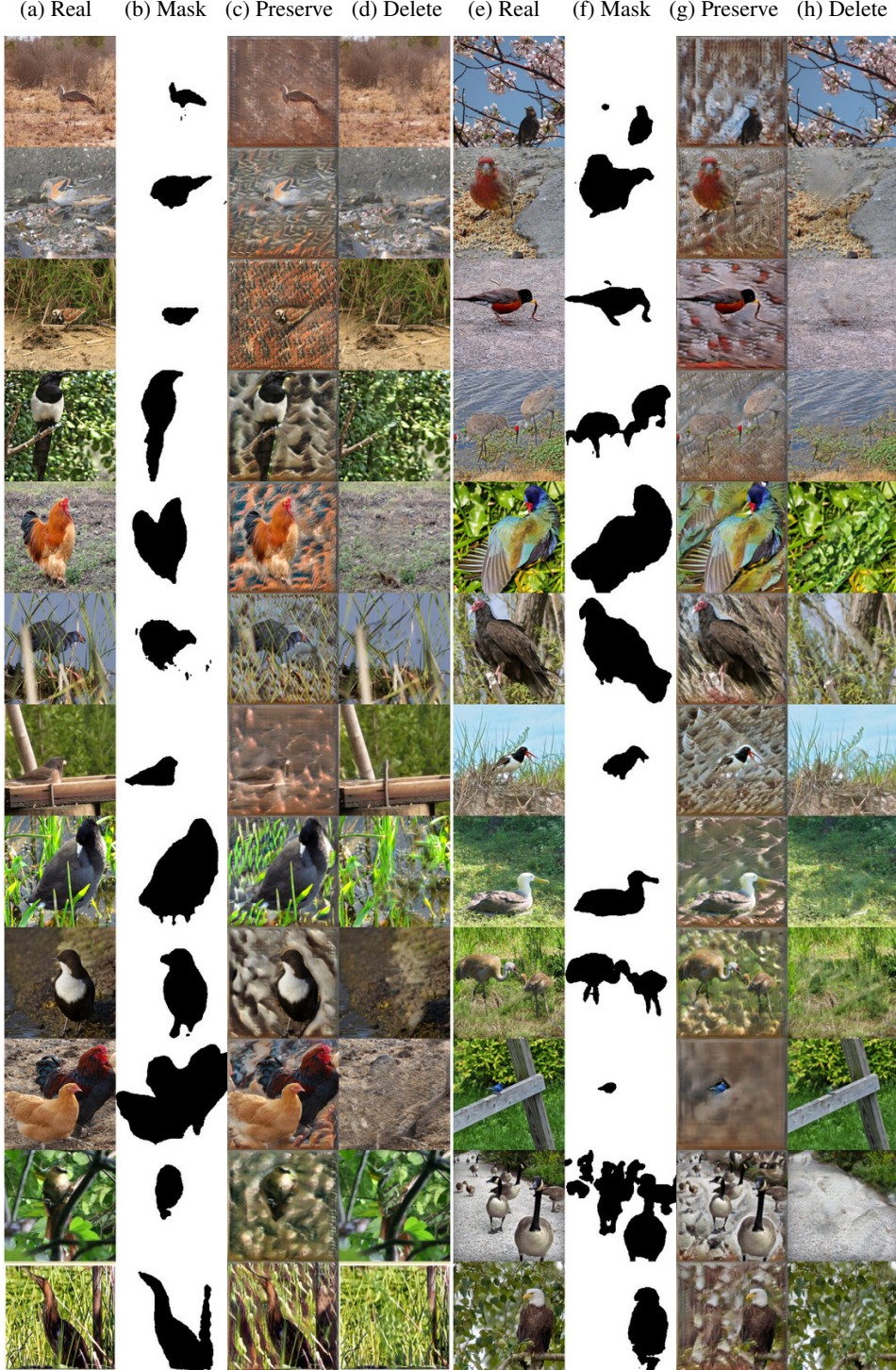

Figure S7: Inpainting using the preservation objective generates unrealistic samples (Sec.4.1). We randomly chose 50 validation-set images (a) from 52 ImageNet bird classes and compute their segmentation masks via a pre-trained DeepLab model (Chen et al., 2017) (b). We found that using the DeepFill-v1 inpainter to inpaint the foreground region (i.e. our "deletion" task) yields realistic samples where the object is removed (d). In contrast, using the inpainter to fill in the background region (i.e. "preservation" task) yields unrealistic images whose backgrounds contain features (e.g. bird feathers or beaks) unnaturally pasted from the object (c).

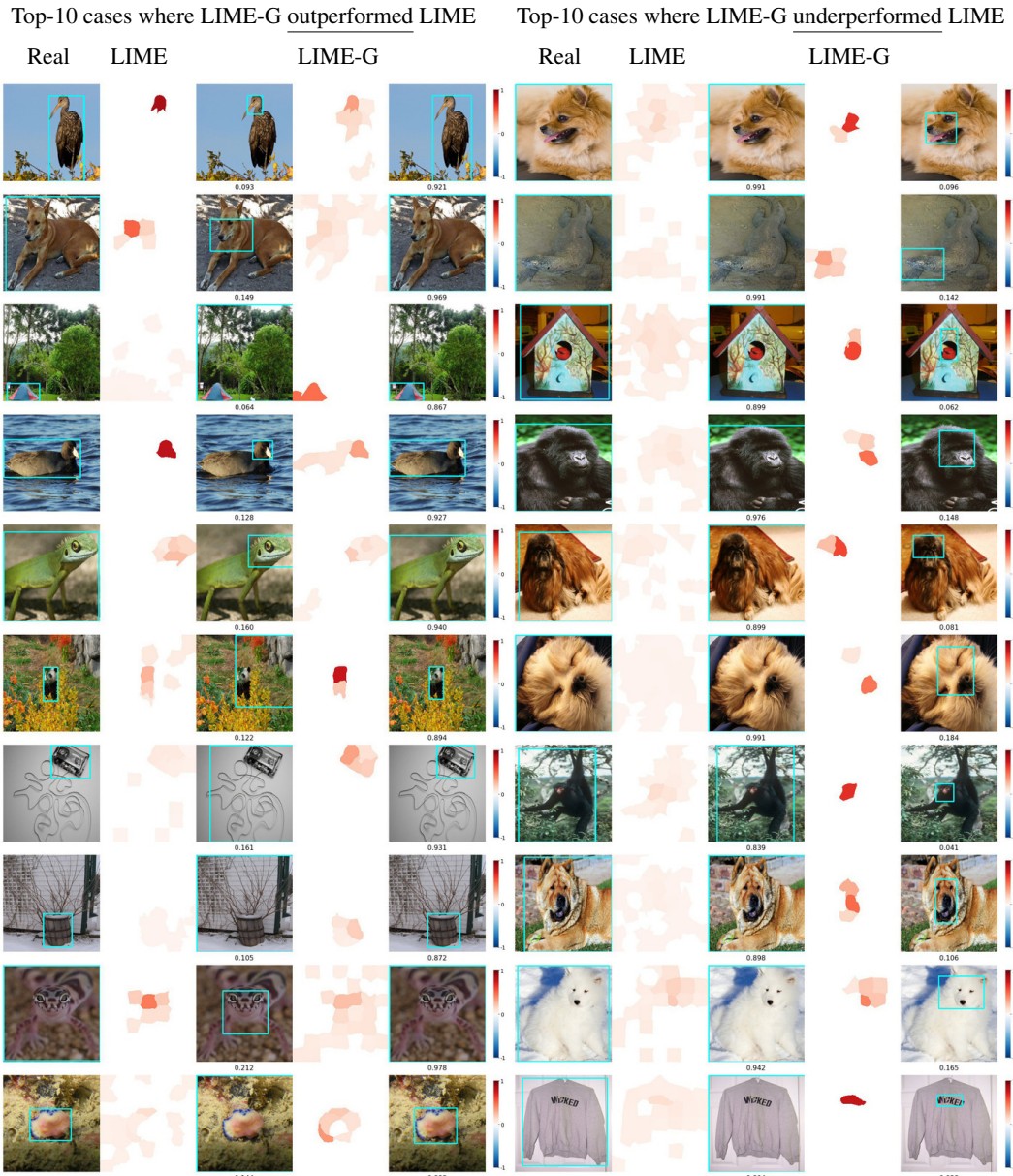

Figure S8: Top-10 cases where the LIME-G outperformed (left) and underperformed (right) LIME on the object localization task (IoU scores). From left to right, on each row: we show a real image with its ground-truth bounding box, LIME heatmap & its derived bounding box, LIME-G heatmap & its derived bounding box. See https://drive.google.com/drive/u/2/folders/10JeP9dpuoa0M16xe2FloBEWajQ7PNKSX for more examples of the LIME and LIME-G IoU results.

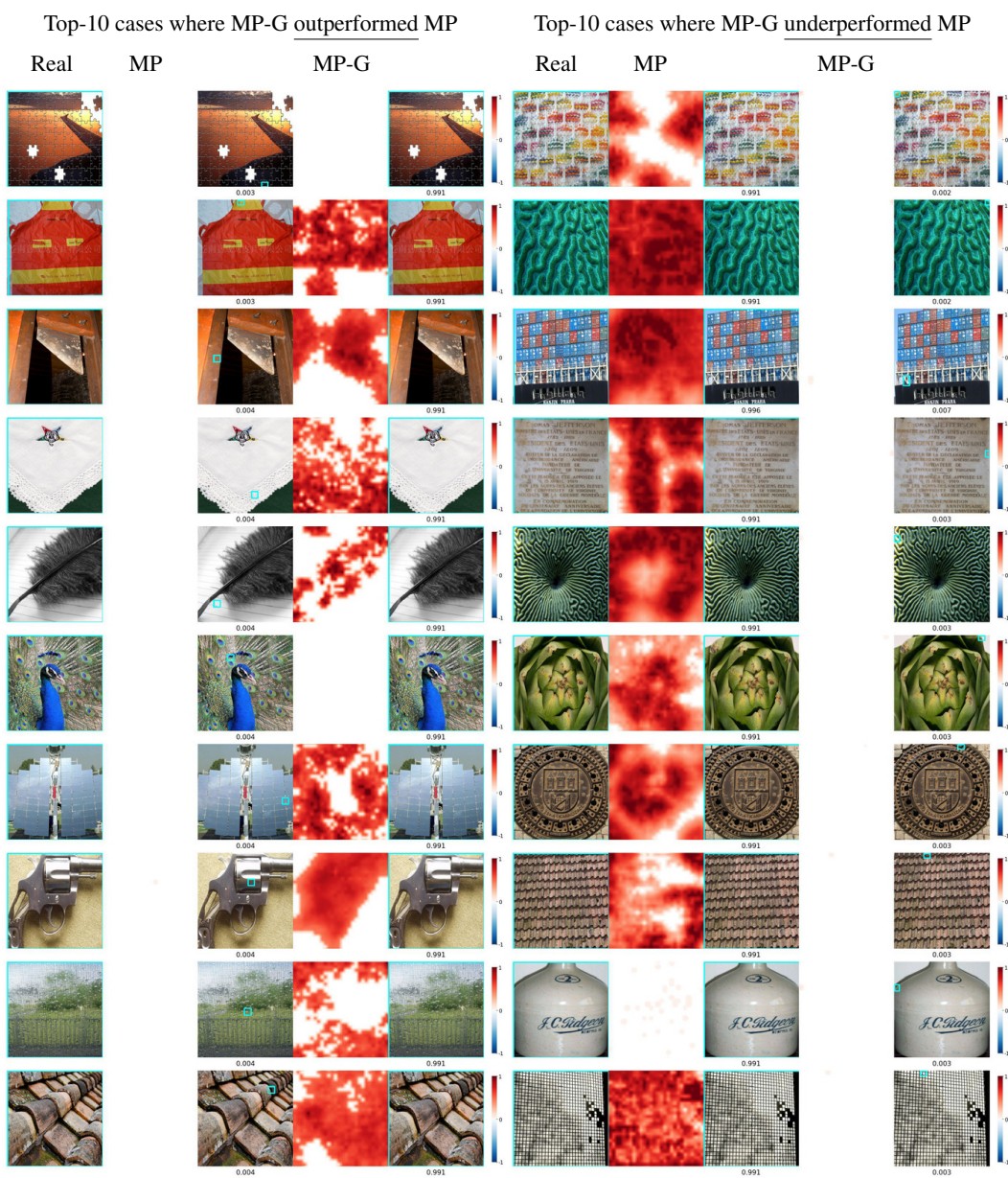

Figure S9: Top-10 cases where the MP-G outperformed (left) and underperformed (right) MP on the object localization task (IoU scores). From left to right, on each row: we show a real image with its ground-truth bounding box, MP heatmap & its derived bounding box, MP-G heatmap & its derived bounding box. See https://drive.google.com/drive/u/2/folders/1nlr5v2RbSiKigp8PRb_aQbauXBmZpHyU for more examples of the MP and MP-G IoU results.

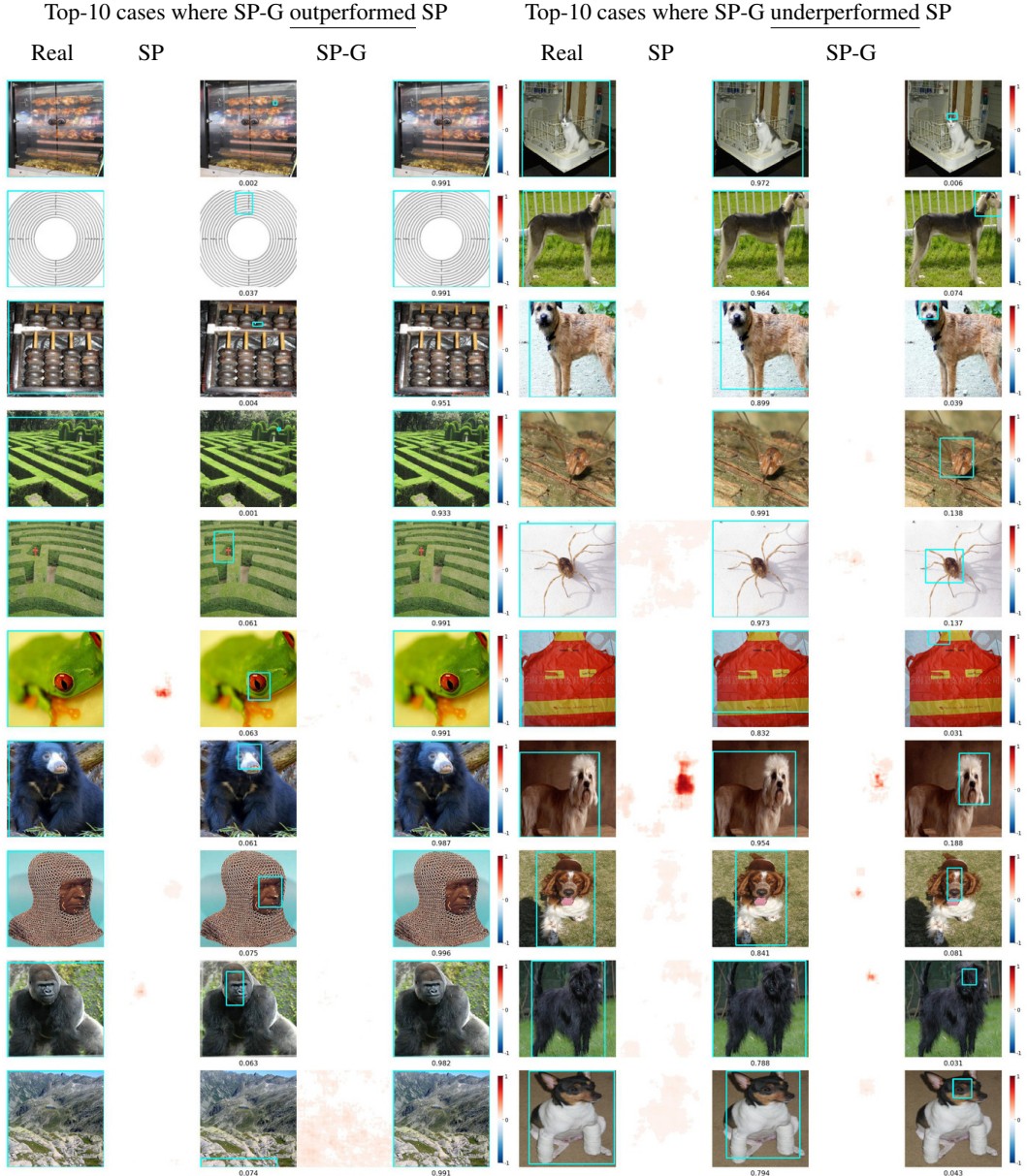

Figure S10: Top-10 cases where the SP-G outperformed (left) and underperformed (right) SP on the object localization task (IoU scores). From left to right, on each row: we show a real image with its ground-truth bounding box, SP heatmap & its derived bounding box, SP-G heatmap & its derived bounding box. In the cases where SP-G has a lower IoU score than SP (right panel), we observed the heatmap localizes some unique features of the object as compared to the images in the top cases where the heatmap covers the entire image. See https://drive.google.com/drive/u/2/folders/1XJ6M0AMHxZrXxLLw6m3Bx7sjvsyqN6JC for more examples of the SP and SP-G IoU results.

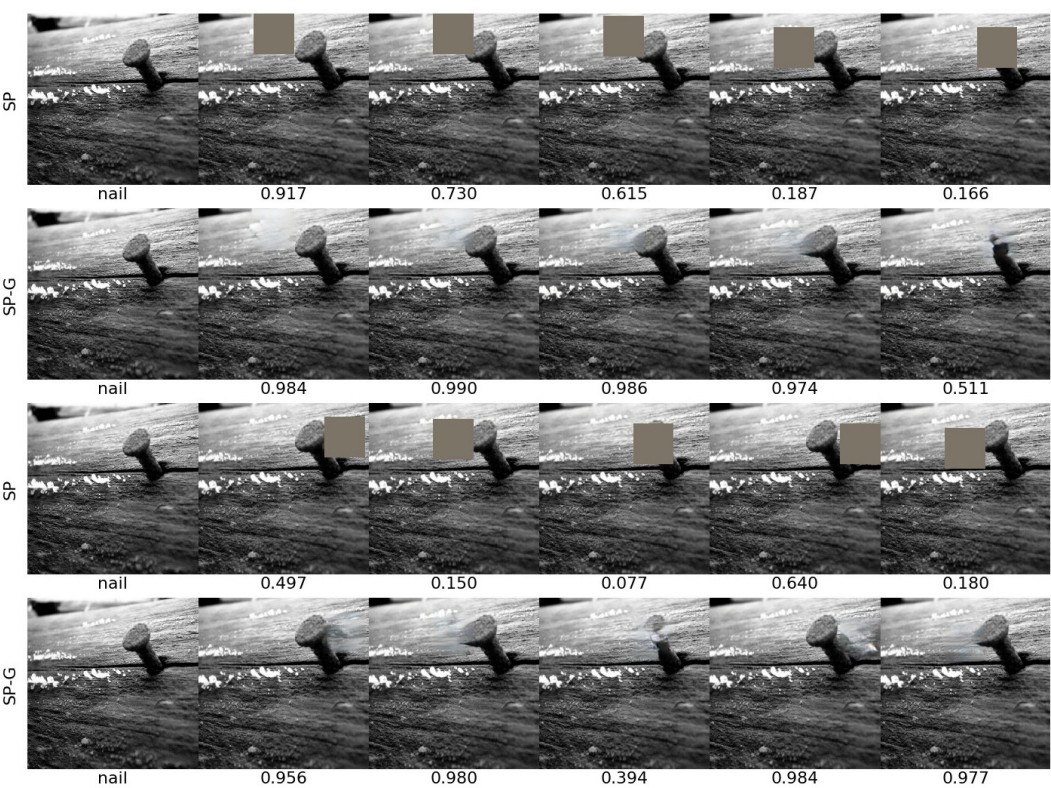

Figure S11: Random intermediate perturbation samples by SP and SP-G on the same image from the nail class in ImageNet. SP-G drops the target-class probability only when the patch cover a major area of the nail (e.g. the center 0.394-probability sample in the bottom panel). This figure is a zoom-in version of the samples in Fig. 4.

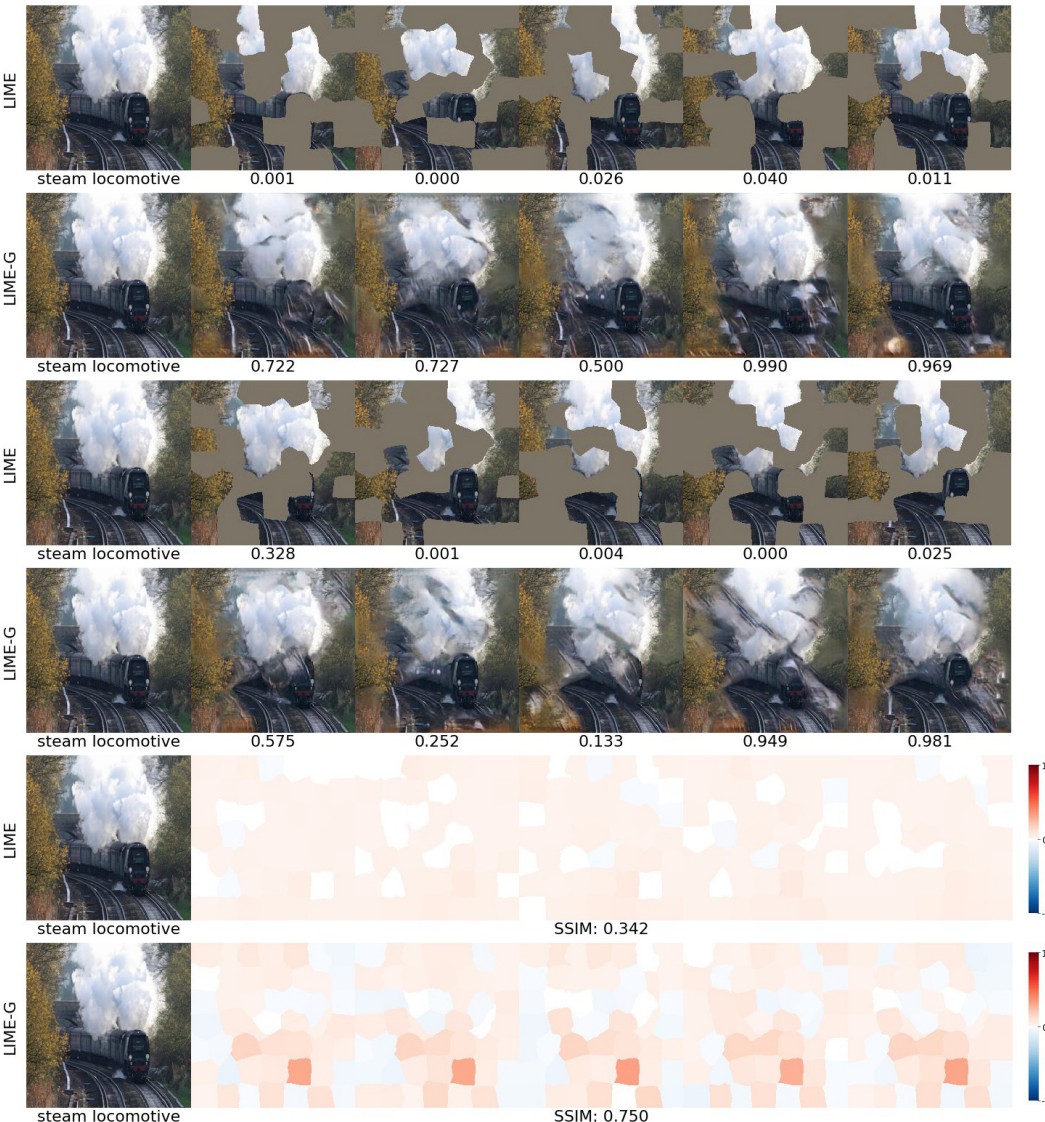

Figure S12: Qualitative evidence supporting the LIME-G vs. LIME sensitivity experiment in Sec. 4.3. For both LIME and LIME-G, per image, we compute an average SSIM score across all 10 pairs of 5 heatmaps. We then take the difference between LIME-G and LIME and sort them in the descending order. This steam locomotive image is a random image from the top-100 ImageNet-S cases where LIME-G outperformed LIME. **Top four rows:** Here, we compare pairs of LIME vs. LIME-G perturbation samples that were created from the same random superpixel masks. LIME-G samples cause large probability drops only when some discriminative feature is removed from the image and thus results in more localized heatmaps. **Bottom two rows:** 5 heatmaps by LIME and LIME-G, each from a random seed. While LIME-G heatmaps are more consistent, LIME heatmaps is noisy and varies. See Fig. S13 and Figs. S14-S15 for similar observations in ImageNet-S and Places365-S dataset respectively. See `https://drive.google.com/drive/u/2/folders/1sKWig4Xk5Pm50kdONdAS9SkiTBhJRAkw` for more examples.

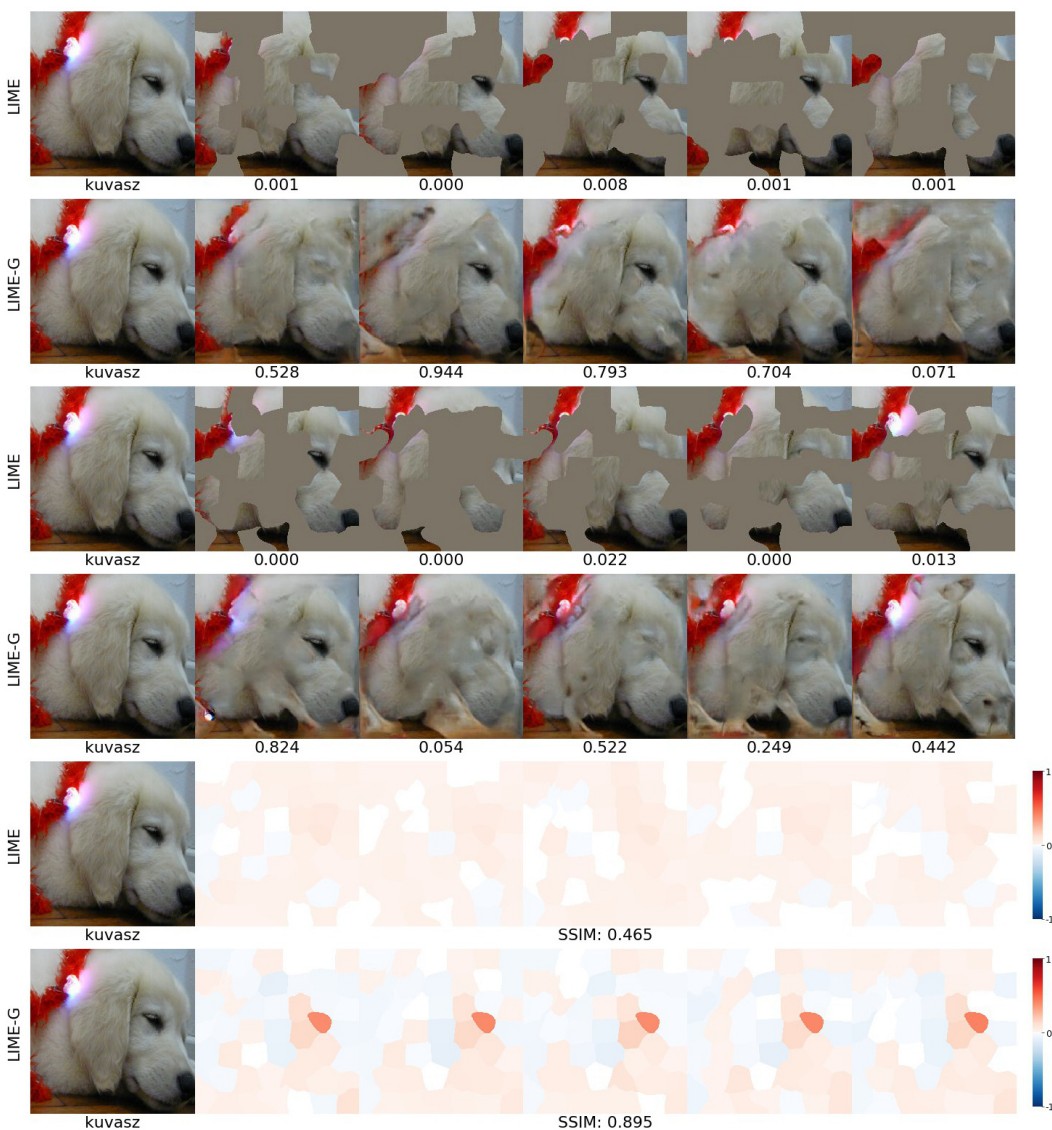

Figure S13: Here, we show the same figure as Fig. S12 (see its caption) but for another random image among the top-100 ImageNet-S cases where LIME-G outperformed LIME on the SSIM similarity metric. See https://drive.google.com/drive/u/2/folders/1sKWig4Xk5Pm50kdONdAS9SkiTBhJRAkw for more examples.

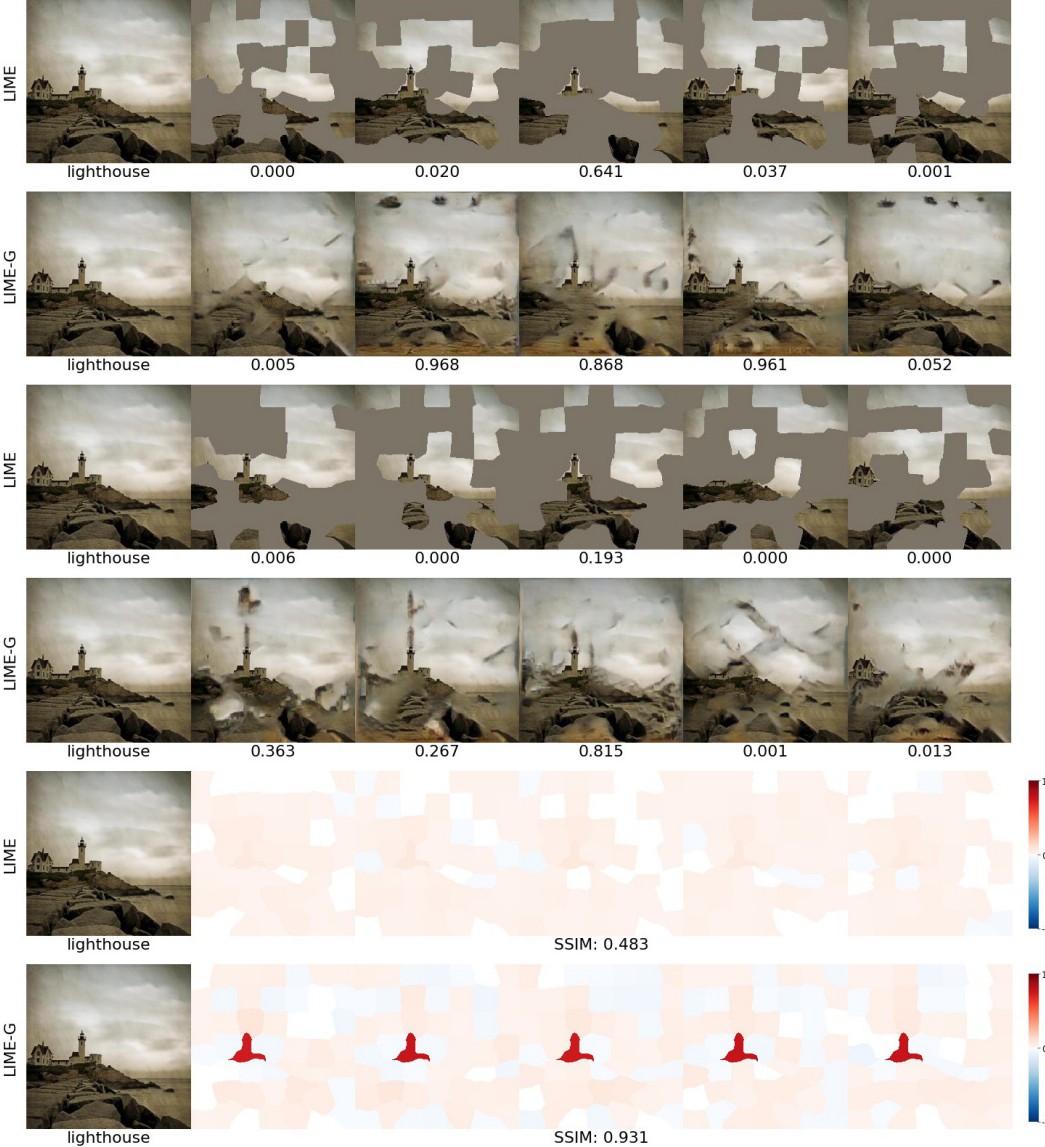

Figure S14: Here, we show the same figure as Fig. S12 (see its caption) but for a random image among the top-100 Places365-S cases where LIME-G outperformed LIME on the SSIM similarity metric. See `https://drive.google.com/drive/u/2/folders/1aXyDFBq0HlcI0kQJpJyspNf2rtwLj35Z` for more examples.

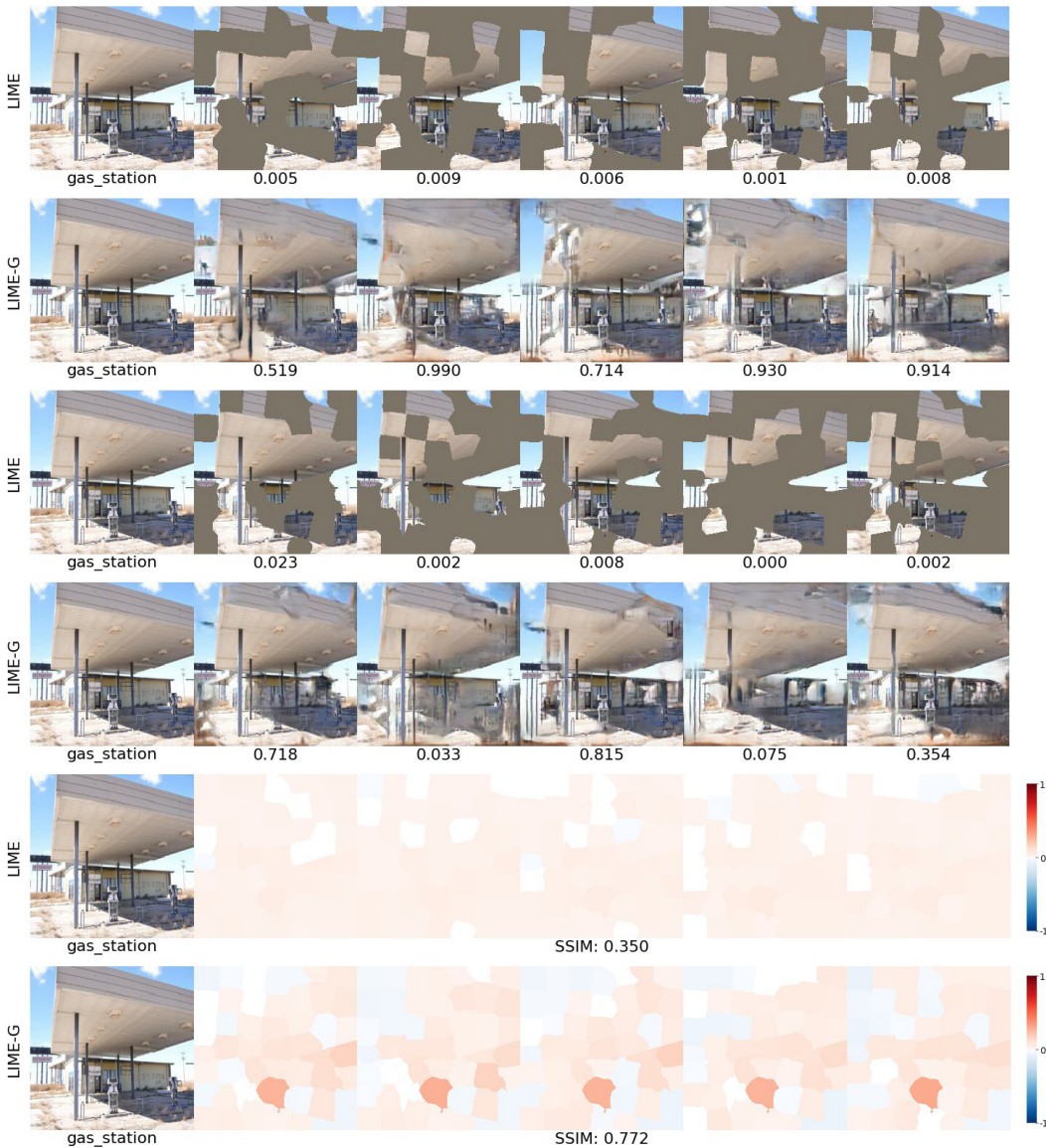

Figure S15: Here, we show the same figure as Fig. S12 (see its caption) but for a random image among the top-100 Places365-S cases where LIME-G outperformed LIME on the SSIM similarity metric. See `https://drive.google.com/drive/u/2/folders/1aXyDFBq0HlcI0kQJpJyspNf2rtwLj35Z` for more examples.

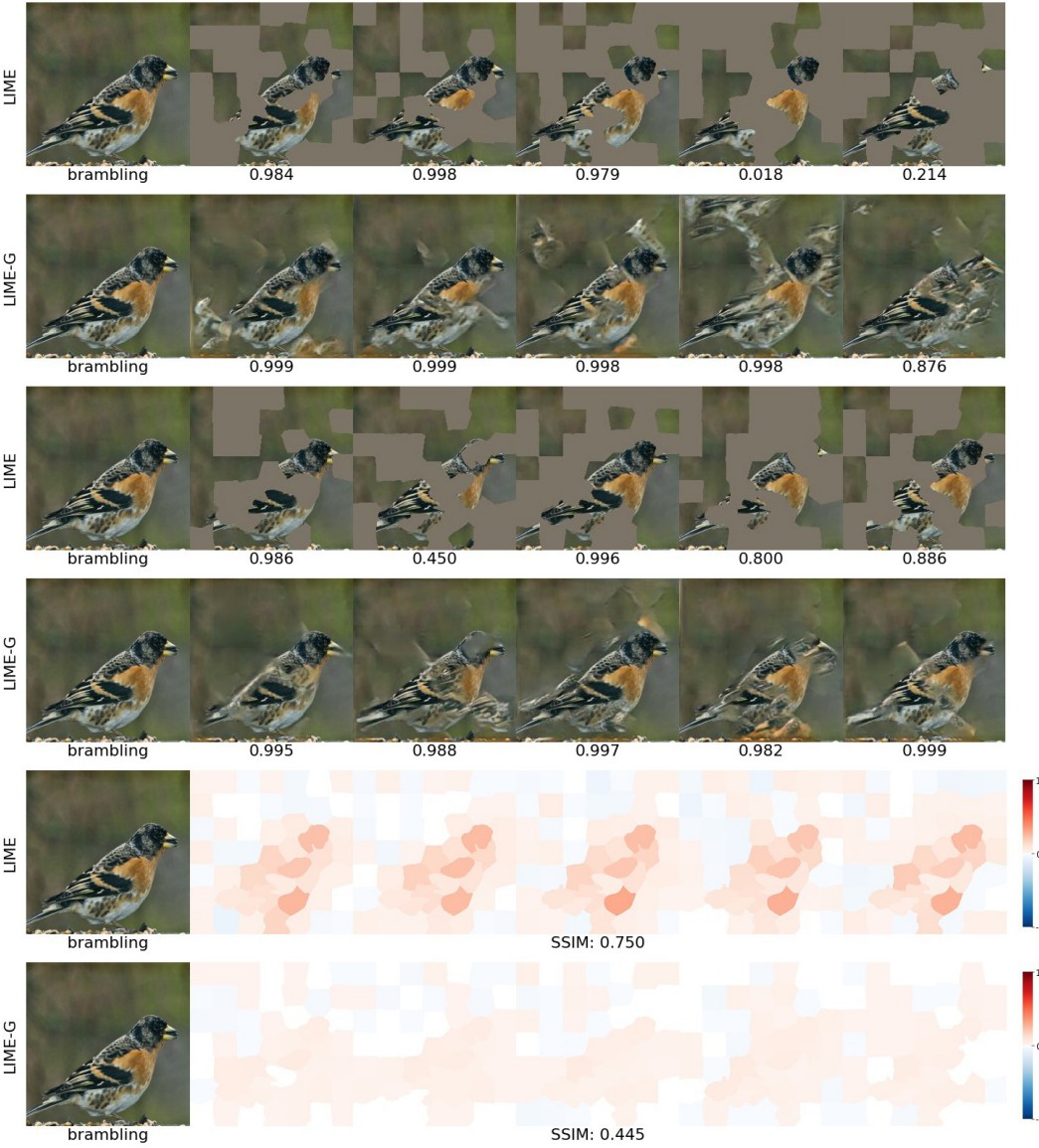

Figure S16: Here, we show the same figure as Fig. S12 (see its caption) but for a random image among the top-100 ImageNet-S cases where LIME-G underperformed LIME on the SSIM similarity metric. LIME-G samples remain at high target-class probabilities and therefore produced heatmaps that are more sensitive than those of LIME. Similar observations can be found in Fig. S17 and Figs. S18-S19. See https://drive.google.com/drive/u/2/folders/1sKWig4Xk5Pm50kdONdAS9SkiTBhJRAkw for more examples.

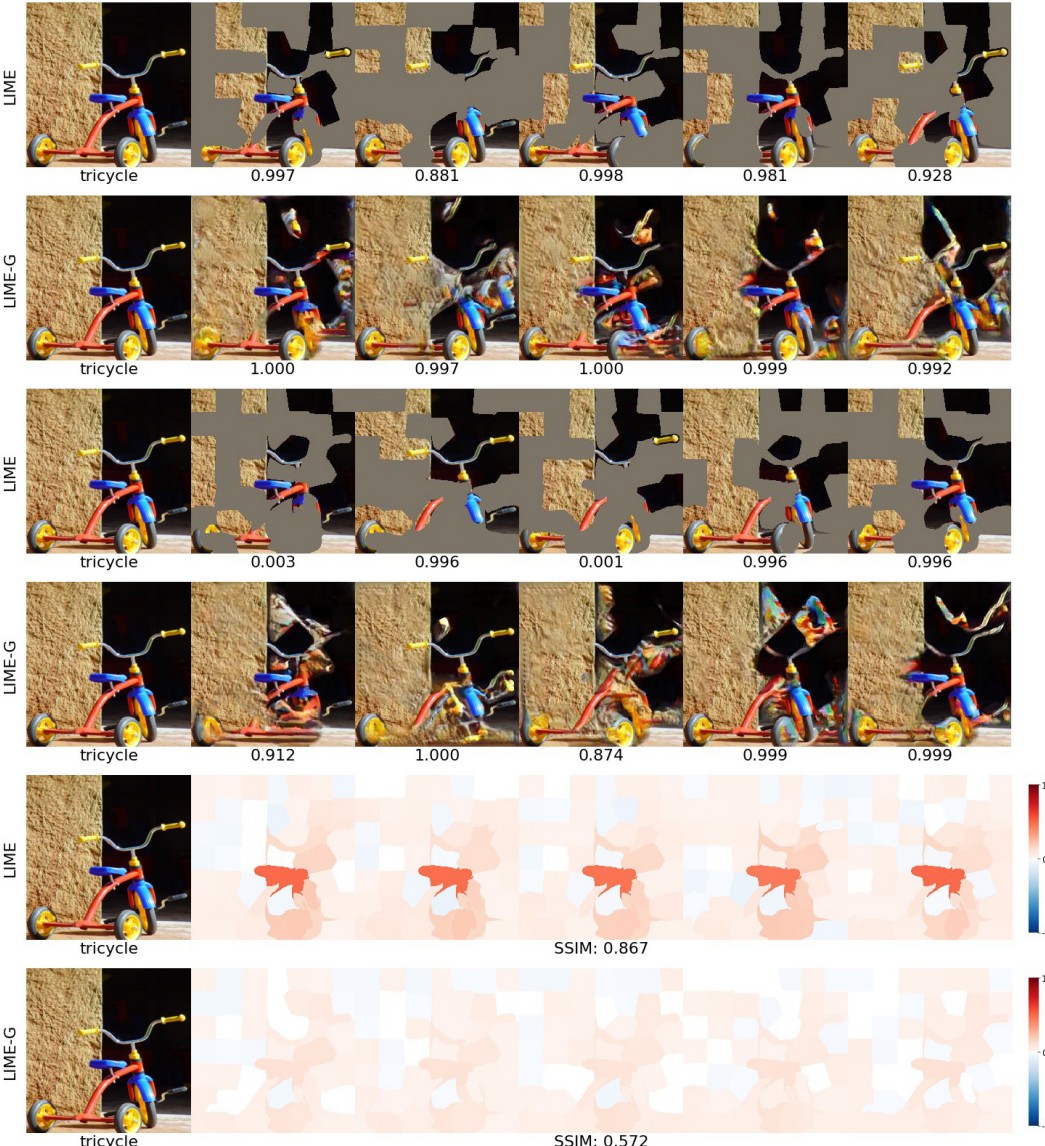

Figure S17: Here, we show the same figure as Fig. S12 (see its caption) but for a random image among the top-100 ImageNet-S cases where LIME-G underperformed LIME on the SSIM similarity metric. See https://drive.google.com/drive/u/2/folders/ 1sKWig4Xk5Pm50kdONdAS9SkiTBhJRAkw for more examples.

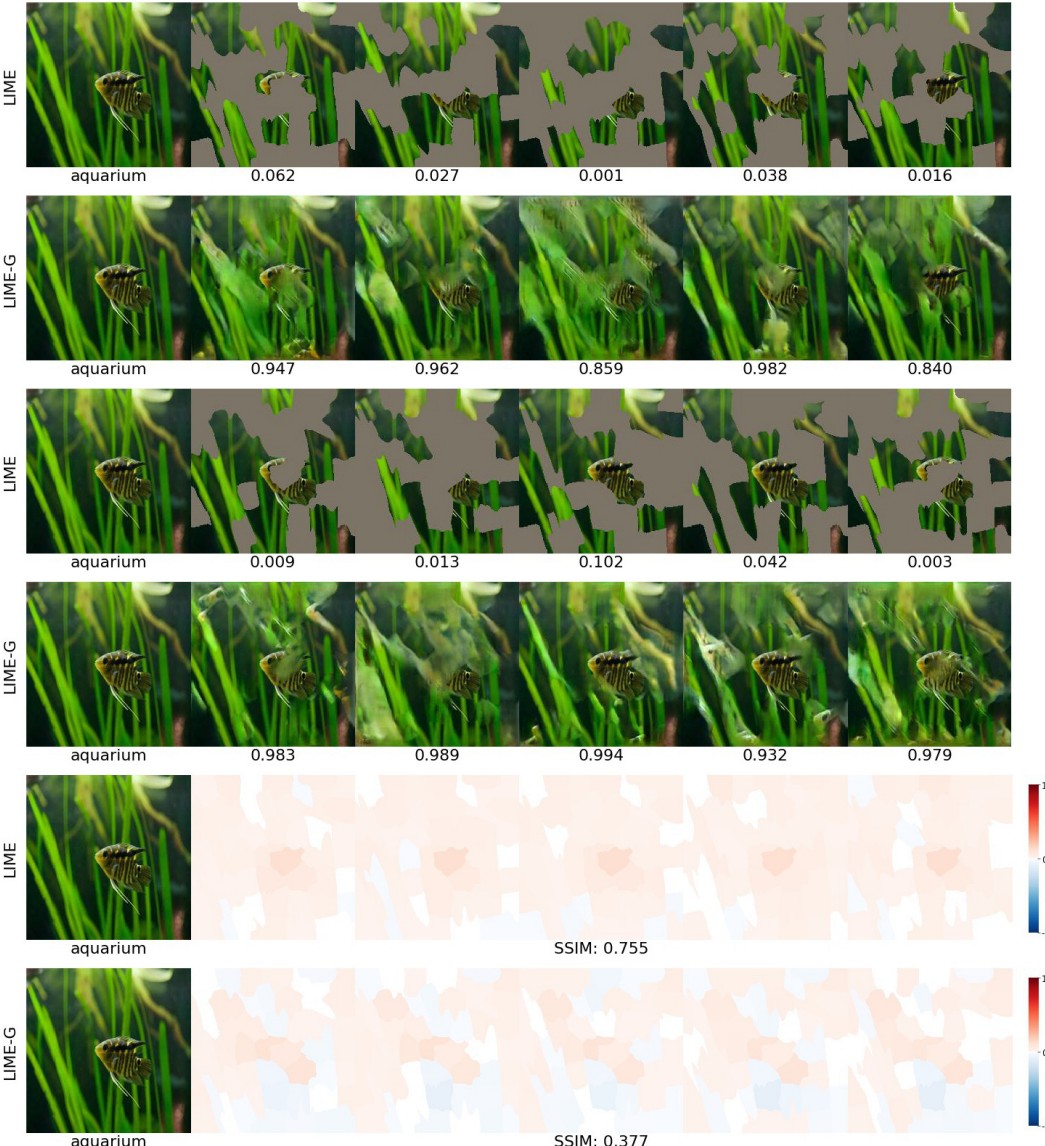

Figure S18: Here, we show the same figure as Fig. S12 (see its caption) but for a random image among the top-100 Places365-S cases where LIME-G underperformed LIME on the SSIM similarity metric. See `https://drive.google.com/drive/u/2/folders/1aXyDFBq0HlcI0kQJpJyspNf2rtwLj35Z` for more examples.

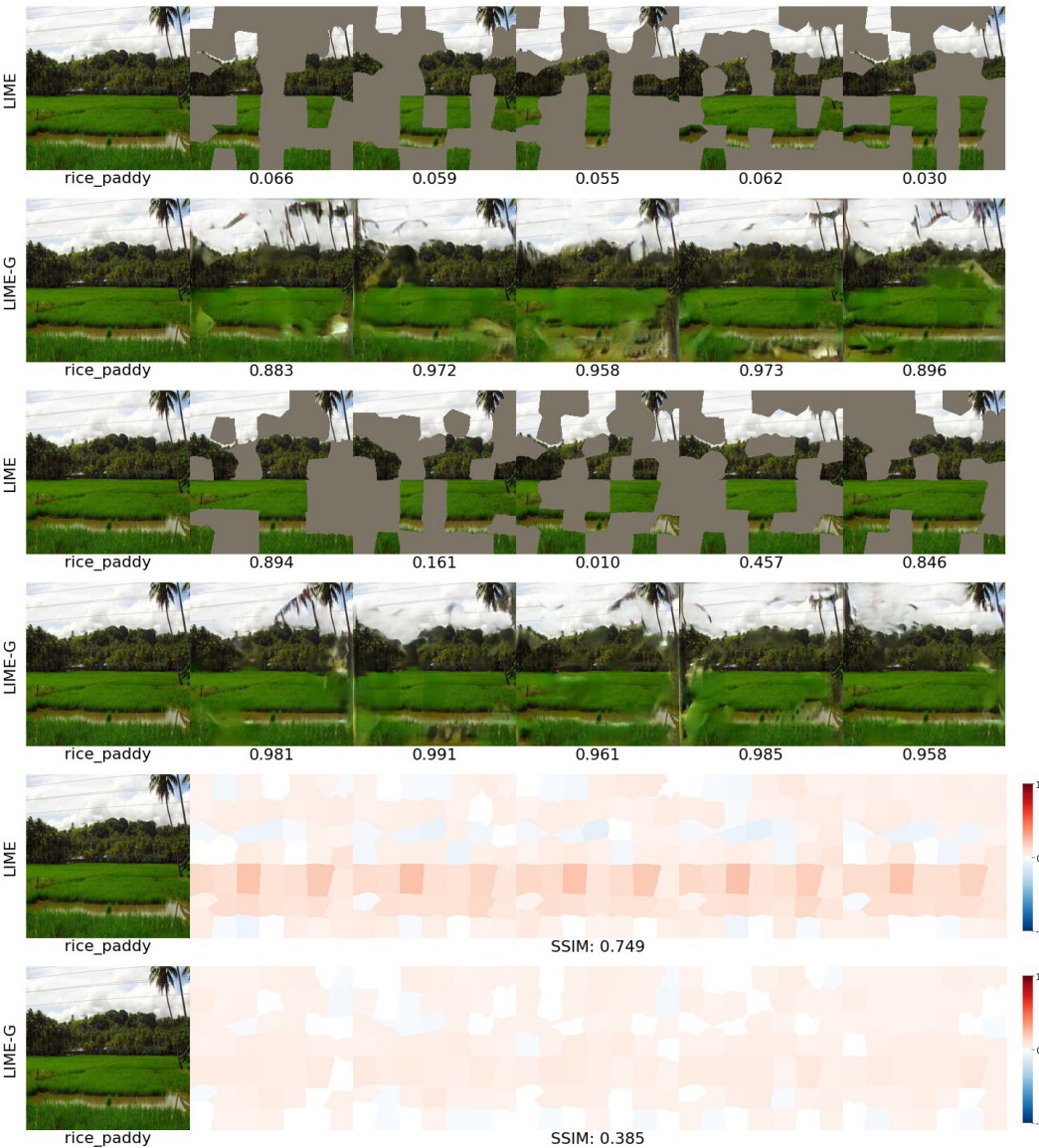

Figure S19: Here, we show the same figure as Fig. S12 (see its caption) but for a random image among the top-100 Places365-S cases where LIME-G underperformed LIME on the SSIM similarity metric. See `https://drive.google.com/drive/u/2/folders/1aXyDFBq0HlcI0kQJpJyspNf2rtwLj35Z` for more examples.

Top-100 cases where LIME-G outperformed LIME   Top-100 cases where LIME-G underperformed LIME

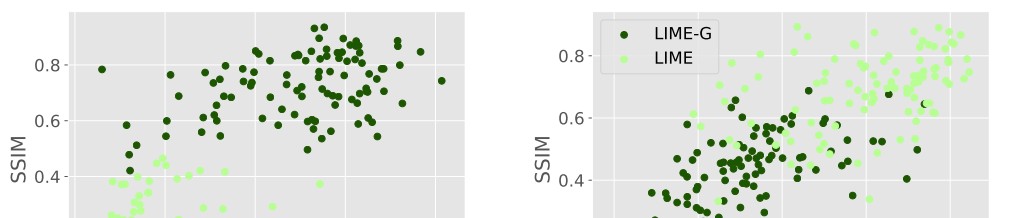

(a) Pearson correlation = 0.856        (b) Pearson correlation = 0.758

Figure S20: Scatter plots for the top-100 images where LIME-G outperformed (a) and underperformed (b) LIME on the SSIM similarity scores. Each data point in the scatter plot we compute two statistics from an image. **y-axis:** We compute the average SSIM score over all 10 possible pairs out of 5 heatmaps computed by running each method (LIME & LIME-G) using 5 random seeds. **x-axis:** We compute the standard deviation (Std.) of the target-class probabilities for all $N = 500$ perturbation samples. We observe a high Pearson correlation between the two measures in both cases when LIME-G outperformed (a) and underperformed LIME (b). That is, the robustness of attribution maps strongly correlates with the variance in the probability distribution of the perturbed samples.

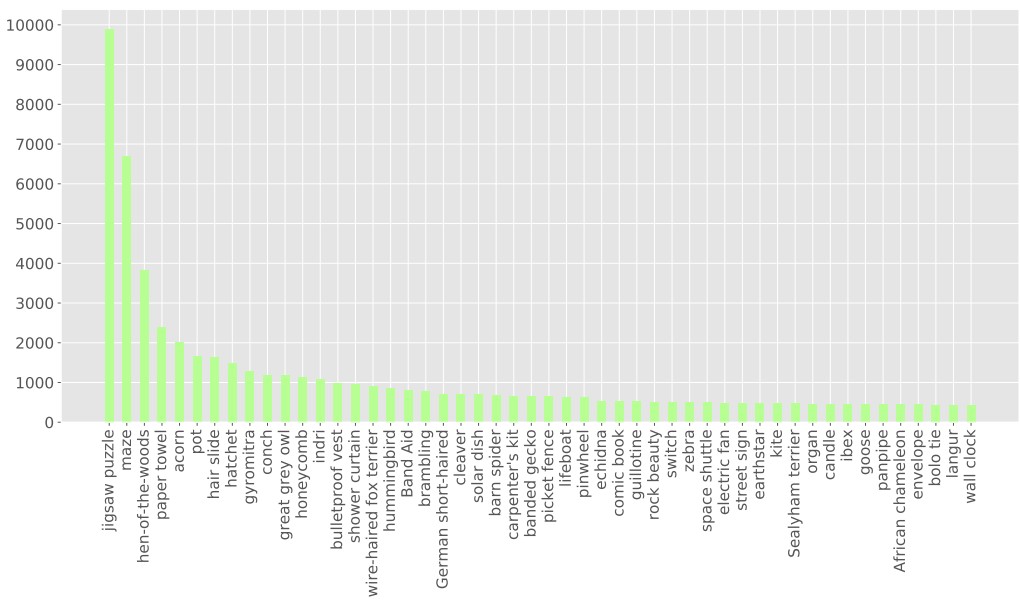

(a) LIME histogram distribution is skewed

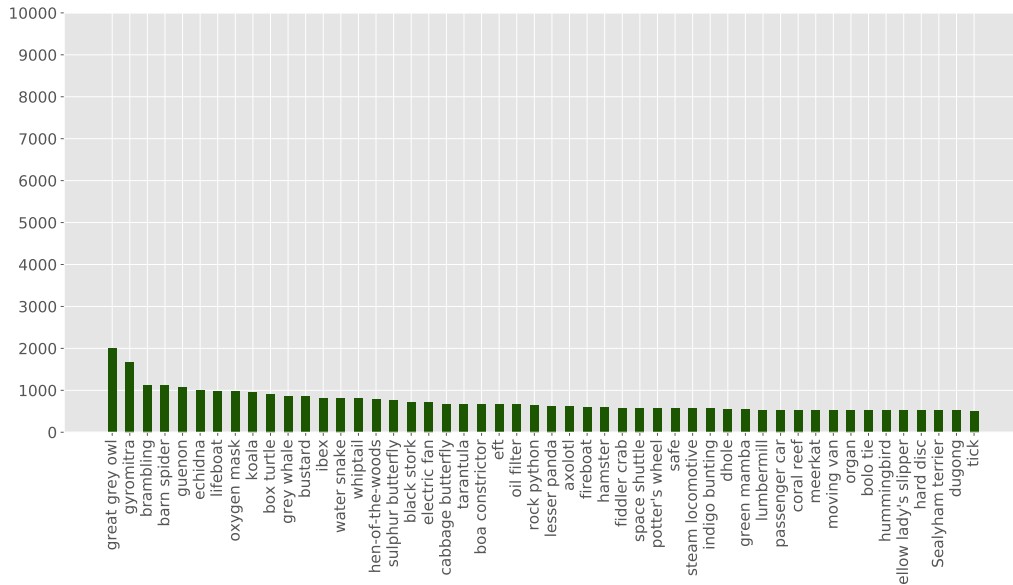

(b) LIME-G histogram distribution is almost uniform

Figure S21: We ran LIME and LIME-G on 200 images, each run has 500 intermediate perturbation samples. Here, for LIME (a) and LIME-G samples (b), we show a histogram of the top-1 predicted class labels for all 200 runs ×500 samples = 100,000 images. The set of 200 images comprises of cases where LIME-G outperformed (100 images) and underperformed (100 images) LIME on the SSIM sensitivity metric (Sec. 4.3). LIME perturbed samples are highly biased towards few jigsaw puzzle, maze classes (top panel), which is somewhat intuitive given the gray-masked images (see Figs. S12–S17). In contrast, the histogram of LIME-G samples are almost uniform. **x-axis:** For visualization purposes, we sorted the top-1 labels and showed only first 50 labels.

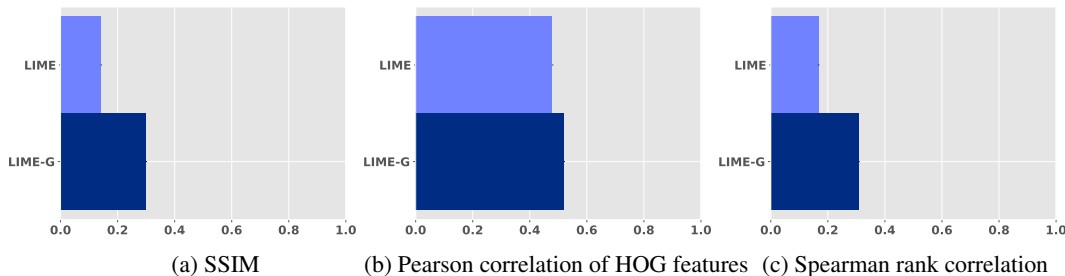

(a) SSIM    (b) Pearson correlation of HOG features (c) Spearman rank correlation

Figure S22: Bar plots comparing the LIME vs. LIME-G robustness (higher is better) across two different numbers of superpixels $S \in \{50, 150\}$ under three different similarity metrics: SSIM (a), Pearson correlation of HOG features (b), and Spearman rank correlation (c). For each image in 1000 random ImageNet-S images, we produced a pair of heatmaps by running LIME (light-blue) or LIME-G (dark-blue) with two different numbers of superpixels $S \in \{50, 150\}$. Each bar shows the mean similarity across all 1000 heatmap pairs. LIME-G is consistently more robust than LIME, specifically by $\sim$200% under the SSIM (a) and Spearman rank correlation (c).

