# OpenReview forum: "Removing input features via a generative model to explain their attributions to classifier's decisions"
_ICLR.cc/2020/Conference — Reject_

### Official Review · AnonReviewer2 · 2019-10-24
**Official Blind Review #2**

**Rating:** 1

**Review:**

The paper proposes a deep visualization technique for black-box image classifiers that feeds modified versions of the original input by means of an off-the-shelf (black box too) image inpainting approach (DeepFill-v1), in order to capture changes in the classification performance. In particular, the substitution of the input image follows three published paradigms: Sliding Patch (SP), Local Interpretable Model-Agnostic Explanations (LIME), Meaningful Perturbation (MP). Whereas the states of the art use gray images (SP, LIME)/blurred versions (MP) as substitution on different spatial supports (regular patch SP, random-shaped superpixel regions LIME, learned continuous region MP), the proposed approach inserts there the output of the inpainting.
There are problems in the paper, major and minor.
Major:
1)	The technical contribute is minimal, the author combine two already existent techniques.
2)	The results are not convincing: a) the comparison are not fair, since on two out of three techniques the authors consistently change the competitors, letting them work with zero-value occluders or random noise. Only the third competitor has been employed in is original form (blurring the images). b) results are better (higher classification drop, plus other metrics) with the proposed approach wrt the first two competitors, which looks strange. Can the authors try to apply the comparative approaches in their original versions?
3)	The fact that we are using an inpainting tool which may work in some cases (in providing well-distributed patches) but in other may fail, corrupting consequently the overall following analysis is a price I don’t want to pay, so I prefer some synthetical but controlled artifact. Actually, in the case of inpainting failure will generate structured noise, hard to be managed.
Minor/improvements
--In the introduction the authors should spend a little more two or three words in explaining on which basis Adebayo et al. 2018 questions the correctness of the heatmap, since this is something on which the authors are building their hypothesis.
--The title is misleading, the authors are talking about generic feature removal but in reality we are considering the image domain only.(check)
--Figure 1’s caption should report the references for the three SP LIME and MP


**Experience Assessment:**

I have published one or two papers in this area.

**Review Assessment: Checking Correctness Of Derivations And Theory:**

I assessed the sensibility of the derivations and theory.

**Review Assessment: Checking Correctness Of Experiments:**

I carefully checked the experiments.

**Review Assessment: Thoroughness In Paper Reading:**

I read the paper thoroughly.

---

> ### Author Response · Authors · 2019-11-07
> **We tried to clarify the main points of our paper**
>
> Thank you for the constructive criticisms and suggestions that helped us revised the paper to be stronger! :)
> We hope to hear your thoughts about our replies below.
>
>
> > The technical contribution is minimal, the author combined two already existent techniques.
>
> We agree and do not claim novel technical contributions. However, we claim to be the first to have studied integrating an inpainter in the correct way into multiple well-known attribution methods, on large-scale image datasets, here, ImageNet and Places365.
>
> We argue that complex theoretical or heavy engineering work is not always necessary or better. Instead, the community should also support simple solutions that move the field in the right direction.
> For example, we would not want to overlook DropOut or ReLUs because of their simple/small technical contributions, right? :)
>
>
> > the comparisons are not fair, since on two out of three techniques the authors consistently change the competitors, letting them work with zero-value occluders or random noise
>
> - We wish to clarify that we did NOT change any original methods here. For LIME we used their own implementation. For MP and SP we implemented ourselves following the original algorithm.
> - The original LIME and SP algorithm remove input pixels by replacing them with 0s.
>
>
> > with the proposed approach wrt the first two competitors, which looks strange. Can the authors try to apply the comparative approaches in their original versions?
>
> We worry that you might have misunderstood the main results. In Sec. 4.2., we did not use SP, LIME, or MP at all. Instead, all we tested is the different filling options that have been used in the literature. The object masks are generated objectively via a third-party DeepLab segmentation network.
>
>
> > The fact that we are using an inpainting tool which may work in some cases (in providing well-distributed patches) but in other may fail, corrupting consequently the overall following analysis is a price I don’t want to pay, so I prefer some synthetical but controlled artifact.
>
> - We agree that generative models are themselves black-boxes. However, we believe that they have reached a performance level that enable them to be useful in synthesizing absences of objects or input features, which is interesting!
> - Importantly, we showed that at the downstream explanation task, the methods using generative inpainters performed better quantitatively—-(1) more robust to hyperparameters; (2) more accurate per localization benchmark.
>
>
> > Elaborating more on the work by Adebayo et al. 2018
> Yes, we are a big fan of their Sanity checks paper! We will try to describe better in a revision.
> We wish to note that Adebayo et al. 2018 studied the heatmap sensitivity to model parameter changes rather than hyperparameter changes (as in our paper)
>
>
> > The title is misleading
> Thank you for the catch! You can see in the revision, we have changed from “classifier” into “image classifier” to address your concern. :)
>
>
> > Fig. 1’s caption should include the references to SP, LIME, and MP
> Yes, we have added the references. Thanks!

---

### Official Review · AnonReviewer4 · 2019-11-01
**Official Blind Review #4**

**Rating:** 3

**Review:**

The paper is focused on perturbation-based local explanation methods; methods that only need black-box(ish) access to the model and generally seek to find a region, pixel, etc's importance score through by removing that region, pixel, etc.. The intuition is that an important region if removed, will result in a large drop in the predicted class confidence. One main issue with such methods is that the removal itself can throw the image out of the data distribution and therefore result in a drop in confidence, not because of the region's importance, but because of the network's unpredictable behavior in such unseen parts of the input space. The work is focused on giving a solution to this problem: instead of removal through blurring, graying, etc, use inpainting; i.e. replace the removed region with using given the rest of the image. The idea has already been discussed out in the literature and the novelty of the work seems to be twofold: They introduce the same method in a way that is not curated for a specific perturbation-based method and could be concatenated with "ANY" given (or future) perturbation-based local explanation method (which authors notate by calling it ${existing_method}-G, they study robustness to hyper-parameter choice.

The paper is quite well written and the experiments are comprehensive. I have two major comments/issues with the work:

1- The contribution of this work given existing work (more specifically the famous Chang et al work seems not to be enough for a venue like ICLR. If I want to list the contributions, it would be as follows (I would appreciate if the authors could correct me as the score is subject to change given more clarification on the matter):
    - This work utilizes an inpainting step in combination with several methods while previous work is focused on meaningful perturbations method. This, although useful, does not introduce a novel technical contribution. The main technical contribution has been the use of inpainting (to be more exact, using generative models to approximate P(c|x_r)) which has been done before on a few previous works.
    - The work argues that the use of inpainting in Chang et al (focused on keeping the salient object and removing background) was invalid as the inpainter model is not trained to do such a task. It could be argued that one could train another inpainting model that "is" capable of such a thing and therefore the general argument would not hold. One drawback of this approach, however, would be that training such an inpainting model might be difficult.
    - Hyperparameter robustness. Studying this question is valuable. However, given that the assumption of this work and previous works is that generative approaches are generally better (even not considering the hyperparameter robustness), I am not sure how this knowledge could be used.

2- Both this work and the previous works run on the assumption mentioned at the beginning of this review which basically says that non-generative perturbation-based methods throw the image out of data distribution and this is bad for such and such reasons. Although intuitively clear, I could not find any evidence in this work suggesting any meaningful difference using objective measures. One would assume that such phenomena would manifest itself clearly using insertion-deletion explanation metrics while as the authors report there was no significant difference. (Section 4.1 results clearly show a difference but this is not related to how the downstream explanation task is affected) For all it is know, generative methods have the drawback of being computationally more expensive than a simple blurring or replacing with random noise. (And a major elephant in the room is whether using an inpainter is actually taking the data back to the true data distribution which seems to be on an unproven assumption that these generative models are capable of learning the data manifold)

Minor comments:
    -Section 4.2 is really interesting. Thanks!
    - Fig 3 results: MP is more robust than MP-G and I couldn't find any explanation of why this method behaves specifically different than the other two in the experiments section. It might be better to move the explanation in the discussions to the experiments.
    - The task of most generative perturbation-based methods is to find a way to approximate P(c|x_\r) which is the conditional probability given the non-removed part of the image. Usually, they do the approximation by sampling several images from the conditional P(x_r|x_\r) (conditional inpainting)  using the generative model and averaging the prediction probability. This work seems not to be concerned with these specifics and directly feeds one of such samples. Could you explain this choice
    - For studying the robustness of LIME, apart from the random seed, couldn't one change the hyperparameters of the superpixel method? Tha one seems more of a practical problem.

**Experience Assessment:**

I have published in this field for several years.

**Review Assessment: Checking Correctness Of Derivations And Theory:**

N/A

**Review Assessment: Checking Correctness Of Experiments:**

I carefully checked the experiments.

**Review Assessment: Thoroughness In Paper Reading:**

I read the paper at least twice and used my best judgement in assessing the paper.

---

> ### Author Response · Authors · 2019-11-07
> **Response to R4 major points**
>
> We really enjoyed your insightful comments a lot! Thank you and we hope to engage you in more discussions here. :-)
> Please find here our responses to your major points.
>
> > Novelty of this work
> We agree that both our work and Chang et al. 2019 (ICLR 2019 paper) used the existing inpainter DeepFill-v1 and therefore we did not claim technical novelty.
> - However, in the interpretability field, we believe the integration of inpainters into attribution methods is an important direction that deserves further research. With a strong generative model of the world, one can ideally generate synthetic intervention samples (here, removing input features) to perform causal inference.
> - We found that the quantitative and qualitative results in Chang et al. 2019 are negative and they did not suggest integrating an inpainter into MP to be a fruitful direction. Their finding is intriguing as it contradicted our intuition and therefore motivated our investigation!
>
>
> > Training an inpainter that is able to remove the background while keeping the salient object
> We agree! Actually, we’d love to work on it in follow-up work because, at the moment, there is no such inpainter in the literature, to the best of our knowledge.
>
>
> > How do generative inpainters help improve the hyperparameter robustness?
> We attempted to provide insights into this excellent question in Sec. 5.2 (which we have also revised to make it clearer. Thanks!).
> The gist is that a strong generative inpainter only allows the classifier probability i.e. P(c|x_\r) to drop when an important discriminative feature is removed, yielding robust/consistent heatmaps upon hyperparameter changes. In contrast, the gray-masked out-of-samples introduced by heuristic perturbations yielded heatmaps that are more noisy and sensitive to even random seed changes (e.g. LIME).
>
>
> > How is downstream explanation result improved with generative models?
> We are too worried about this question and therefore have attempted all common evaluation metrics, which have their own pros/cons. We found our methods outperformed the counterparts on two objective metrics:
>
> - On the well-known Localization task (Sec. 4.4), which have been used in many influential papers [1-6] (including Chang et al. 2019), SP-G and LIME-G consistently outperformed SP and LIME. The assumption of the Localization task is that better explanations should highlight the salient object in an image, which is imperfect but arguably reasonable for the ImageNet dataset which is object-centric.
> - On our sensitivity evaluation (Sec. 4.3), which is critical for ensuring good attribution methods, SP-G and LIME-G also consistently outperformed SP and LIME on both ImageNet and Places365 datasets.
> - We are aware of the recent BAM evaluation dataset by Google Brain, which appeared on arXiv (https://arxiv.org/pdf/1907.09701.pdf) recently. We are excited to test our methods on it soon!
>
> [1] Fong, R. C., & Vedaldi, A. (2017). Interpretable explanations of black boxes by meaningful perturbation. In Proceedings of the IEEE International Conference on Computer Vision (pp. 3429-3437).
> [2] Wagner, J., Kohler, J. M., Gindele, T., Hetzel, L., Wiedemer, J. T., & Behnke, S. (2019). Interpretable and fine-grained visual explanations for convolutional neural networks. In Proceedings of the IEEE Conference on Computer Vision and Pattern Recognition (pp. 9097-9107).
> [3] Zhou, B., Khosla, A., Lapedriza, A., Oliva, A., & Torralba, A. (2016). Learning deep features for discriminative localization. In Proceedings of the IEEE conference on computer vision and pattern recognition (pp. 2921-2929).
> [4] Chang, C. H., Creager, E., Goldenberg, A., & Duvenaud, D. (2018). Explaining image classifiers by counterfactual generation, ICLR, 2019.
> [5] Selvaraju, R. R., Cogswell, M., Das, A., Vedantam, R., Parikh, D., & Batra, D. (2017). Grad-cam: Visual explanations from deep networks via gradient-based localization. In Proceedings of the IEEE International Conference on Computer Vision (pp. 618-626).
> [6] Zhang, J., Bargal, S. A., Lin, Z., Brandt, J., Shen, X., & Sclaroff, S. (2018). Top-down neural attention by excitation backprop. International Journal of Computer Vision, 126(10), 1084-1102.
>
>
> > Does an inpainter is actually taking the data back to the true data distribution?
> We agree that this is an important note. However, it could be argued that the same doubt exists for all generative tasks in Machine Learning. That is, density estimation for high-dimensional data is challenging and evaluating generative models is an active open research (famously as seen in the GAN or NLP literature).
>
>
> We will reply to your minor points in another reply.

---

> > ### Comment · AnonReviewer4 · 2019-11-15
> > **Rebuttal**
> >
> > Thank you very much for your detailed response and for addressing all of the questions. Unfortunately, I was not convinced that the contributions of the work are enough for the ICLR venue. The general direction of the work, however, points towards a promising solution to the existing methods' drawbacks.

---

> > > ### Author Response · Authors · 2019-11-15
> > > **Could you elaborate on your reasons?**
> > >
> > > Thank you so much for taking the time to respond to us!
> > >
> > > We would greatly appreciate it if you could elaborate on your reasons for "contributions are not enough for ICLR".
> > > We really wish to improve the manuscript further in light of your comments.
> > >
> > > re: results
> > > - If you worry about the insignificant result of Insertion & Deletion (despite our argument that this metric has its own problems), we'd be happy to include other results of Saliency Metrics [1] and Pointing Game [2].
> > >
> > > On these two metrics, our preliminary tests are showing that G methods are superior to the original methods (similar conclusion as for the well-known Localization task that we used in the paper).
> > >
> > > - We'd be happy to also try other metrics if you suggest to provide more evidence! :)
> > >
> > > [1] Dabkowski, Piotr, and Yarin Gal. "Real time image saliency for black box classifiers." Advances in Neural Information Processing Systems. 2017.
> > > [2] Zhang, Jianming, et al. "Top-down neural attention by excitation backprop." International Journal of Computer Vision 126.10 (2018): 1084-1102.
> > >
> > > re: novelty
> > > - We think Chang et al. 2019 results are misleading, and completely opposite to what we claim in this paper.
> > > It is a pity for the community if work that does things in the correct way but being second is discouraged.

---

> ### Author Response · Authors · 2019-11-07
> **Response to R4 minor points**
>
> Please see below our replies to your other comments (which helped us a lot in revising the paper!).
>
> > Move the MP explanation in the discussions to the experiments.
> Thanks for your suggestion! We will make this change in a revision.
>
>
> > How about sampling more images from the conditional P(x_r|x_\r) and averaging the prediction probability?
> This is honestly what we had wanted to do from the beginning of the project i.e. marginalizing over all possible scenarios where a feature is absent. However, state-of-the-art image inpainting methods are currently only able to synthesize *one single* output per input image. There were recent efforts in CVPR 2019 in the face domain e.g. [1]. However, on ImageNet or Places365, none of them were really working in practice (confirmed by our own preliminary tests and correspondence with the authors).
>
> That is, we did not want to add more complex theoretical bits that do not really work yet in practice. The best and publicly available inpainter we could find is DeepFill-v1 (used in this paper).
>
> [1] Zheng, Chuanxia, Tat-Jen Cham, and Jianfei Cai. "Pluralistic Image Completion." Proceedings of the IEEE Conference on Computer Vision and Pattern Recognition. 2019.
>
>
> > Apart from the random seeds, how about testing LIME sensitivity to the number of superpixels?
> We tested the sensitivity to changing random seeds is because that sensitivity is unavoidable to everyone when running LIME due to the stochastic sampling of the random masks.
> But thanks to your suggestion! We have also ran statistics for that and found that LIME-G is consistently more robust than LIME (on all 3 similarity metrics: SSIM, Pearson, Spearman) when we vary the number of superpixels between { 50, 150 }.
>
> See the plot here https://drive.google.com/open?id=1W5jWz-3nshiT0tL6BlE4I6hc5xjpweNk  ---LIME-G heatmaps are always more consistent than those of LIME (dark bars are longer than light bars). This figure is also now included in a revision.

---

### Official Review · AnonReviewer3 · 2019-11-05
**Official Blind Review #3**

**Rating:** 6

**Review:**

The paper proposes to improve perturbation-based explanation techniques by complementing the perturbation step with an inpainting step that removes artefacts caused by it.

The approach is sound and intuitive.

The authors show the flexibility of their approach by applying it to a variety of perturbation-based attribution techniques.

It is unclear what is the computational cost of the inpainter. Perturbation-based explanations are generally quite slow due to having to evaluate the function many times, and therefore a further slowdown could harm practical use. I'm curious whether the inpaiting approach could be in some way also extended to faster explanation techniques (e.g. gradient-based, or propagation-based).

Evaluation experiments are not fully conclusive. Bounding box experiments are rather indirect and the deletion/insertion metrics do not systematically show the performance improvement of using inpainting. Perhaps the deletion metric should have been equiped with inpainting as well in order to avoid deletion artefacts. (See e.g. Samek'17 MoRF / LeRF experiments where various perturbation schemes are tested for deletion).

The evaluation benchmark is restricted to perturbation-based approaches. It could have been useful to broaden the comparison to non-perturbation approaches.

An experiment I found particularly interesting is the robustness to perturbation hyperparameters. Given the difficulty of designing evaluation metrics that can support hyperparameter selection, hyperparameter insensitivity is indeed strongly desirable.

I'm wondering whether it is really necessary to use a strong deep neural network inpainter since the goal is just to remove artefacts. Some inpainters provided as part of standard computer vision libraries work quite well, and do not need to be trained and adapted to a certain shape of missing data.

Overall, the paper presents an interesting and sound approach to improve perturbation-based explanations. Experiments are extensive, although some of them remain so far not fully conclusive.

**Experience Assessment:**

I have published in this field for several years.

**Review Assessment: Checking Correctness Of Derivations And Theory:**

N/A

**Review Assessment: Checking Correctness Of Experiments:**

I assessed the sensibility of the experiments.

**Review Assessment: Thoroughness In Paper Reading:**

I read the paper at least twice and used my best judgement in assessing the paper.

---

> ### Author Response · Authors · 2019-11-07
> **Response to R3 comments 1/2**
>
> > It is unclear what is the computational cost of the inpainter.
>
> Thank you for the note! We will add it in a revision.
>
> - It takes us 0.019 secs to inpaint one 256x256 image (01 forward pass) using 1 GTX 1080Ti.
> - 01 MP vs. MP-G optimization run of 300-steps to generate one attribution map:
> MP: 7.273 secs / run
> MP-G: 13.530 secs / run
>
> - 01 LIME vs. LIME-G run (50 superpixels, 1000 samples) to generate one attribution map:
> LIME: 2.222 secs / run
> LIME-G: 10.779 secs / run (we obtained this empirical number yesterday for a batch of 10; however, in theory it should be ~4.2 secs / run).
>
> Note that for LIME or SP, one can speed up a run further by using larger batch sizes.
>
>
> > Could the inpainting approach be also extended to faster explanation techniques (e.g. gradient-based, or propagation-based)?
>
> We are not sure to understand the suggestion. Our approach is applicable to any method that attempts to remove an input feature. Fast attribution methods such as LRP, Input x Gradient or vanilla Gradient attempt to construct a heatmap analytically and therefore, do not perturb the input image in the first place (thus, we do not see inpainting applicable there).
>
>
> > Bounding box experiments are rather indirect and the deletion/insertion metrics do not systematically show the performance improvement.
>
> Thanks for your note! Evaluation is indeed what we worry about a lot because every attribution map evaluation metric (including Insertion/Deletion) is imperfect with its own pros and cons.
>
> - On (1) our sensitivity evaluation and (2) Localization task (which is widely used in the literature), SP-G and LIME-G consistently outperformed SP and LIME on both ImageNet and Places365. So the findings here are significant.
>
> - For Insertion / Deletion, the results are mixed. For example, both ImageNet and Places365, SP-G outperformed SP under Deletion, but not Insertion. In contrast, LIME-G outperformed LIME under Insertion, but not Deletion.
>
> - For honest reporting, we avoid drawing conclusions here. Note that for the Insertion / Deletion metrics has a drawback: the number of pixels to be zero-ed at once is also a hyperparameter to tune, which may change the entire conclusion!
> We are running our evaluation on this hyperparameter and will report in a revision.
>
> - Per our literature survey, the Insertion and Deletion metrics are less widely used than the Localization task. In contrast to our paper, previous work often only reports either Insertion e.g. in [1] or only Deletion e.g. in [2], but did not compare whether the conclusions generalize on both metrics.
>
> [1] Samek, W., Binder, A., Montavon, G., Lapuschkin, S., & Müller, K. R. (2016). Evaluating the visualization of what a deep neural network has learned.
> [2] Wagner, J., Kohler, J. M., Gindele, T., Hetzel, L., Wiedemer, J. T., & Behnke, S. (2019). Interpretable and fine-grained visual explanations for convolutional neural networks. CVPR.
>
>
> > Perhaps the deletion metric should have been equipped with inpainting?
>
> We have cited Samek et al. 2017 in a revision for clarity. Thanks for a nice pointer and an interesting idea!
> We will try to run the suggested evaluation and report it here.
>
>
> > How about broadening the evaluation benchmark to non-perturbation approaches?
>
> We appreciate your suggestion but we do not see the purpose of evaluating non-perturbation attribution methods in the context of our paper. Our approach here is merely to improve the perturbation-based techniques. That is, we test replacing heuristic perturbation approaches by a learned perturbation method, i.e. inpainting, that creates more plausible samples.
>
> Please let us know if you think otherwise. We’d be happy to consider any suggested evaluation!
>
>
>
> > An experiment I found particularly interesting is the robustness to perturbation hyperparameters.
>
> Thank you, we think it is an important evaluation metric moving forward as well!

---

> ### Author Response · Authors · 2019-11-07
> **Response to R3: DeepFill-v1 is the best choice regarding both inpainting speed and image quality**
>
> > Is it necessary to use a strong generative inpainter?
>
> That’s a great question!
> a) Short answer: we indeed had tried other non-learning approaches but we found the learned inpainter DeepFill-v1 to be more preferable due to both (1) significantly faster inpainting speed; (2) arguably better quality (see quantitative comparisons in [1]).
>
> [1] Yu, J., Lin, Z., Yang, J., Shen, X., Lu, X., & Huang, T. S. (2018). Generative image inpainting with contextual attention. In Proceedings of the IEEE Conference on Computer Vision and Pattern Recognition (pp. 5505-5514).
>
> b) Long answer: We had tried using PatchMatch (https://gfx.cs.princeton.edu/pubs/Barnes_2009_PAR/patchmatch.pdf), a state-of-the-art out-of-the-box non-learning inpainting approach. However, PatchMatch is actually an extremely slow, iterative method that compares arbitrary pairs of patches in the same image to find the closest match to fill in part of the missing region.
> - It takes an average of 472.35 secs / image to inpaint using PatchMatch using multithreading on CPU. In contrast, a forward pass through DeepFill-v1 takes only 1.5 seconds per image on CPU.
> - Image evaluation is subjective by nature, therefore, we also include a qualitative comparison between PatchMatch vs. DeepFill-v1, in case you are interested.
> https://drive.google.com/open?id=1pAwSJGS5llOsh2INDnuhoDprFkrZJXwP (note that PatchMatch left many blank areas for some cases because it failed to find matching patches within 100 iterations—-a hyperparameter.

---

### Author Response · Authors · 2019-11-06
**[ To all reviewers ] The contributions of this work**

We would like to thank all three reviewers for their positive feedback and insightful questions! We have revised the paper in light of many of your comments.

We wish to clarify the contributions of our work in this common thread and will address each reviewer’s comments separately in other replies.


1. Our contribution to the interpretability community

The three attribution methods being studied in our work are all well-known and have ~10,000 citations in total i.e. SP by Zeiler et al., 2014 (7521 cites), LIME by Ribeiro et al., 2016 (1991 cites), and MP by Fong & Vedaldi, 2017 (246 cites).
These famous papers have and will continue to inform our interpretability community i.e. ~40,000 papers on Google Scholar that contain both the “interpretability” and “machine learning” keywords.

However, we found two worrisome issues among them and many other attribution methods (including the related Chang et al. 2019): (1) the perturbed samples are clearly unrealistic; (2) some methods are so sensitive to hyperparameter changes.

For example, we found that 20.5% of LIME perturbation samples are labeled by ResNet-50 into one of the three classes: jigsaw puzzle, maze, and hen-of-the-wood. That is, these grayed-out masked images look mostly like puzzles or mazes to the classifier (regardless of the actual input image). A histogram of labels of LIME samples is in Fig. S21 in the revision (also here, https://drive.google.com/file/d/1-p8AedfD-5dHjco4ZlAVHIe9jLPOccu1/view?usp=sharing).

We are the first to show that, for SP and LIME, ameliorating such an issue of “unrealistic intervention samples” has consistently produced more (1) accurate attribution maps (by the Localization task) and (2) more robust attribution maps.
We believe our work is important to the community and the contrary negative result with MP-G interestingly suggests a need for better a inpainter in the future.


2. Our contribution regarding attribution map evaluation metrics

We agree and are deeply worried that the interpretability field is currently lacking a good, commonly accepted benchmark.

--2.a) The localization task has been well-known and used in many influential and also recent interpretability papers. On this task, our SP-G and LIME-G are consistently superior to SP and LIME.

--2.b) Our insignificant result on the Insertion / Deletion metrics does not necessarily imply G-methods are not promising. Because these two metrics are based on the assumption that input pixels are *independent* and are better-suited for evaluating fine-grained attribution maps e.g. Gradient or Integrated Gradient, but not the coarse methods such as SP, LIME or MP.

That is, there are many open issues: (1) Insertion / Deletion evaluation requires zero-ing out input pixels, and therefore yield adversarial examples—-the exact issue we attempt to solve in this paper; (2) how many pixels should be knocked out at one?; (3) if the pixel-independence assumption is unrealistic, should we knock out superpixels instead?

--2.c) Due to the lack of good evaluation metrics, we proposed to also evaluate attribution methods by their sensitivity to hyperparameters. To the best of our knowledge, this important factor is often neglected in prior work and we are the first to throughly evaluate heatmaps both previous (SP, LIME, MP) and ours (SP-G, LIME-G, MP-G) under this sensitivity.


3. Our contribution given the previous work by Chang et al. 2019

- Chang et al. 2019 (ICLR 2019 paper) was the first to apply the inpainter, but on *only* one attribution method which is the Meaningful Perturbation (MP). However, qualitatively, their inpainted samples are largely unrealistic due to the unsuitable use of the “Preservation” objective function, defeating the key purpose behind using an inpainter in the context of removing input features to evaluate their importance.
- Importantly, they also evaluated on the Localization task as we do here. And their quantitative results were negative i.e. they found the integration of an inpainter produced worse localization performance. They did not evaluate on the Insertion or Deletion metrics. Therefore, per demonstration of Chang et al. 2019, there was NO quantitative benefits of using an inpainter.
-  In contrast, we found in our paper that integrating the inpainter is indeed a promising direction for two *different* methods: Sliding Patch (SP) and LIME. That is, our methods LIME-G and SP-G are consistently, on both ImageNet and Places365, better in both metrics (1) Localization; and (2) Robustness to hyperparameter changes.
- Our finding that MP-G is inferior to MP is somewhat consistent with the finding in Chang et al. 2019. Our result is actually interesting given that we used a different objective function (i.e. Deletion) and might inform future research (our hypothesis: MP operates at the pixel level and requires an inpainter that is able to perform free-form inpainting but DeepFill-v1 is not good at that task).

Thank you!

---

### Decision · Program_Chairs · 2019-12-19

**Decision:**

Reject

**Comment:**

Perturbation-based methods often produce artefacts that make the perturbed samples less realistic. This paper proposes to corrects this through use of an inpainter.  Authors claim that this results in more plausible perturbed samples and produces methods more robust to hyperparameter settings.
Reviewers found the work intuitive and well-motivated, well-written, and the experiments comprehensive.
However they also had concerns about minimal novelty and unfair experimental comparisons, as well as inconclusive results. Authors response have not sufficiently addressed these concerns.
Therefore, we recommend rejection.